



# Neural networks in catchment hydrology: A comparative study of different algorithms in an ensemble of ungauged basins in Germany

Max Weißenborn[1], Lutz Breuer[1,2], and Tobias Houska[1]

[1] Institute for Landscape Ecology and Resources Management (ILR), Research Centre for BioSystems, Land Use and Nutrition (IFZ), Justus Liebig University Giessen, Heinrich-Buff-Ring 26, 35390 Giessen, Germany
[2] Centre for International Development and Environmental Research (ZEU), Justus Liebig University Giessen, Senckenbergstraße 3, 35392 Giessen, Germany

**Correspondence:** Max Weißenborn (maxweissenborn.umwelt@gmail.com)

**Abstract** This study presents a comparative analysis of different neural network models, including Convolutional Neural Networks (CNN), Long Short-Term Memory (LSTM) and Gated Recurrent Unit (GRU) in predicting discharge within ungauged basins in Hesse, Germany. All models were trained on 54 catchments with 28 years of daily meteorological data, either including or excluding 11 static catchment attributes. The training process of each model scenario combination was repeated

100 times, using a Latin Hyper Cube Sampler for the purpose of hyperparameter optimisation with batch sizes of 256 and 2048. The evaluation was carried out using data from 35 additional catchments (6 years) to ensure predictions in basins that were not part of the training data. This evaluation assesses predictive accuracy, computational efficiency concerning varying batch sizes and input configurations and conducted a sensitivity analysis of various hydrological and meteorological. The findings indicate that all examined artificial neural networks demonstrate significant predictive capabilities, with a CNN model

exhibiting slightly superior performance, closely followed by LSTM and GRU models. The integration of static features was found to improve performance across all models, highlighting the importance of feature selection. Furthermore, models utilising larger batch sizes displayed reduced performance. The analysis of computational efficiency revealed that a GRU model is 41% faster than the CNN and 59% faster than the LSTM model. Despite a modest disparity in performance among the models (<3.9%), the GRU model's advantageous computational speed renders it an optimal compromise between predictive accuracy

and computational demand.



# 1 Introduction

Artificial intelligence (AI) is increasingly being used to answer scientific questions, including those in the realm of hydrology (Kratzert et al., 2019a, b; Afzaal et al., 2019; Nabipour et al., 2020). The predictive accuracy of AI in these hydrological studies, particularly concerning discharge, is of paramount importance for flood control, watershed management or the estimation of

water availability (Sharma and Machiwal, 2021; Brunner et al., 2021). In the era of climate change, which causes tremendous variability in rainfall patterns and increases evapotranspiration, the role of precise hydrological forecasts becomes even more essential (Tabari, 2020). An area of particular challenge is prediction in ungauged basins (PUB), an endeavour fraught with substantial uncertainty due to the lack of empirical data for model calibration (Blöschl, 2016). Effective models for PUB should thus possess robust generalisation capabilities across diverse watershed behaviours, enabling more universal basin-type

predictions (Sivapalan et al., 2003).

As demonstrated by Kratzert et al. (2019a), an artificial neural network (ANN) model, namely Long Short–Term Memory (LSTM) model, has shown unprecedented accuracy in PUB. The employed LSTM model exhibited the ability to generalise rainfall–runoff predictions across a substantial number of basins (531), surpassing the performance of traditional hydrological models that typically operate best when independently calibrated for each separate basin.

Further comparative analyses, such as those by Le et al. (2023), have evaluated the performance of LSTM against other ANNs like multilayer perceptrons (MLP) and convolutional neural networks (CNN) in daily streamflow prediction. This study revealed superior performance of LSTM and CNN models over conventional ANNs, with LSTM exhibiting a marginal edge over CNN. Moreover, a novel approach proposed by Ghimire et al. (2021) involves a hybrid CNN-LSTM model, designed for hourly discharge predictions. When benchmarked against various ANNs (CNN, LSTM, DNN), traditional AI models (Extreme

Learning Machine, MLP), and ensemble methods (Decision Tree, Gradient Boosting Regression, Extreme Gradient Boosting, Multivariate Adaptive Regression Splines), the CNN-LSTM model displayed superior performance in multiple evaluation metrics, although all ANNs exhibited high efficacy. This evidences that deep learning, a subset of machine learning characterised by multilayered ANNs, holds substantial promise for streamflow prediction. However, while numerous studies have explored discharge prediction using ANNs, a limited number have conducted comparative analyses of different ANN architectures. Ta-

ble 1 summarises these studies from 2020 to December 2023, noting that most incorporate lagged target variables as inputs. This methodology, though effective, is less applicable for PUB due to the absence of discharge data in ungauged or poorly gauged regions, necessitating the use of discharge–independent inputs. Among the studies shown in Table 1, three specifically address this constraint. The first, by Nguyen et al. (2023a), evaluates CNN and LSTM models for daily discharge prediction in the 3S River Basin, exclusively using daily mean temperature and precipitation data. This study adopted a "one model fits all"

approach, akin to Kratzert et al. (2019a), training both model architectures with data from all three sub-basins. The LSTM was found to outperform the CNN, although the latter's results were not extensively discussed. The second study, by Wegayehu and Muluneh (2023), contrasts three super ensemble learners against eight base models, including LSTM, Gated Recurrent Unit model (GRU), and a compound CNN-GRU model, for daily discharge prediction. Here, the LSTM ranked among the





top three in four out of five scenarios based on $R^2$ metrics. However, its performance significantly declined in the absence of
feature selection, indicating a susceptibility to redundant features. Notably, this study trained separate models for each basin,
thus not directly addressing PUB generalisation capabilities. The third study, by Oliveira et al. (2023), compared three ANN
models (LSTM, CNN, and MLP) for daily discharge estimation in a single basin. The CNN model exhibited superior performance (NSE of 0.86); however, this does not imply generalisability in non-calibrated catchments as both calibration and testing
occurred within the same basin. Regrettably, this limitation pertains to all three studies.

Consequently, this research aims to bridge the existing literature gap by comparing the performance of three distinct ANN
architectures for predicting discharge in ungauged basins. Through a comparative analysis, this study not only addresses a significant gap in hydrological literature but also provides valuable insights into the relative strengths and limitations of each ANN
model, thereby guiding future applications and development in the field of hydrological prediction. Furthermore, a comprehensive sensitivity analysis was conducted to identify key drivers affecting the prediction of each model. This methodological
approach contributes to refining model selection and calibration strategies in hydrological forecasting.

The first architecture under examination is the LSTM, which has demonstrated robust performance in numerous studies
(Kratzert et al., 2019a, b; Le et al., 2023; Nguyen et al., 2023a). Although LSTM models display promising performance,
the sequential architecture of the LSTM incurs at tremendous computational costs, resulting in a relative lag in computational
efficiency when contrasted with feed–forward neural networks or CNN (Gauch et al., 2021). In pursuit of addressing these
limitations and challenges inherent to LSTM models, the second architecture chosen for examination is the CNN. This model
is distinguished by its parallel processing capabilities, significantly boosting computational efficiency, a critical factor when
handling large-scale, high-resolution time series data, extensive input sequences, and a multitude of input features (Bai et al.,
2018). The third architecture under consideration is the Gated Recurrent Unit. GRU, a variant of LSTM, is renowned for its
proficiency in effectively capturing temporal dependencies in time series data while imposing less computational burden (Cho
et al., 2014).

Given that PUB is often characterised by data scarcity this study incorporates two distinct scenarios: the first involving the
use of only daily forcing data, and the second extending this with additional static catchment features. This approach allows
for an evaluation of the model's generalisation capacity when constrained to minimal data. Additionally, it provides insights
into the degree to which static catchment features can contribute to enhancing model performance, as indicated by (Kratzert
et al., 2019a). Accordingly, the objectives of this study are delineated as follows:

  i. to evaluate the potential of predicting discharge in ungauged basins by daily forcing data with ANNs, namely LSTM,
     CNN, and GRU,

  ii. to compare the computational efficiency of LSTM, CNN, and GRU models for daily time series prediction,

  iii. to investigate the potential of static features to enhance prediction performance, and

  iv. to assess the impact of batch size on model performance and computational efficiency.



**Table 1.** Overview of recent studies focused on comparing discharge prediction using various artificial neural networks. 'Target independence' indicates that discharge data were not utilised as input features during model training/testing. 'Ungauged' implies model evaluation with catchments, that were not part of the training dataset. 'Multi catchment' denotes that the models were evaluated on multiple catchments. ANFIS=Adaptive neuro-fuzzy inference system; ANN=Artificial neural network; BiLSTM=Bidirectional LSTM; CNN=Convolutinal neural network; DT=Decision tree; DTR=Decision tree regressor; FNN=Feedforward neural network; GB=Gradient boosting; GRU=Gated recurrent unit; LSTM=Long short-term memory; LR=Linear regression; MLP=Multilayer perceptron; LASSO=Least absolute shrinkage and selection operator; PSO=Particle swarm optimization; Res=Residual; RF=Random forest; RNN=Recurrent neural network; SVR=Support vector regression; XGB=Extreme gradient boosting

| Target independent | Ungauged | Multi catchment | Time scale | Lead time step Single | Multi | Prediction algorithm | Reference |
|:---:|:---:|:---:|:---|:---:|:---:|:---|:---|
| ✔ | ✔ | ✔ | Daily | ✔ | | CNN, GRU, LSTM | This study |
| ✔ | | ✔ | Daily, Monthly | ✔ | | CNN, LSTM | Nguyen et al. (2023a)[a] |
| ✔ | | | Daily | ✔ | | CNN-GRU, GRU, LR, LSTM, LASSO, MLP, SVR, XGB | Wegayehu and Muluneh (2023)[b] |
| ✔ | | | Daily | ✔ | | CNN, LSTM, MLP | Oliveira et al. (2023) |
| | | ✔ | Daily | ✔ | ✔ | CNN, LSTM, MLP, Transformer | Nguyen et al. (2023b) |
| | | ✔ | Daily, Monthly | ✔ | ✔ | ANN, LSTM | Cheng et al. (2020)[b] |
| | | | Daily | ✔ | | ANFIS, ANN, BiLSTM, CNN-GRU-LSTM | Vatanchi et al. (2023)[b] |
| | | | Daily | ✔ | | ANN, CNN, LSTM | Le et al. (2023) |
| | | | Daily | ✔ | | ANFIS, LSTM-PSO | Haznedar et al. (2023)[b] |
| | | | Daily | ✔ | | CNN-LSTM, DT, GB, LSTM, MLP, RF | Hong et al. (2020)[b] |
| | | | Daily | ✔ | ✔ | BiLSTM, CNN, FNN, GRU, LSTM, StackedLSTM | Le et al. (2021) |
| | | | Daily | ✔ | | CNN, DTR, LSTM, RF | Li et al. (2022)[b] |
| | | | Daily | ✔ | | CNN-LSTM, DT, GB, MLP, RF, RNN-LSTM | Hong et al. (2021)[b] |
| | | | Daily | | ✔ | CNN-LSTM, LSTM | Deng et al. (2022)[b] |
| | | | Daily | | ✔ | BiLSTM, CNN-LSTM, ResBiLSTM, ResCNN-LSTM | Herbert et al. (2021) |

[a] Only results of LSTM model is stated, [b] hyperparamter configuration nontransparent





## 2 Materials and Methods

### 2.1 Study Area

All basins analysed in this study are located in the federal state of Hesse, Germany (Figure 1). The climate of this region is temperate–humid and characterised by moderate temperature and precipitation levels (Heitkamp et al., 2020). The topography of Hesse, characterised by a complex blend of lowlands, hilly terrains and modest mountain ranges, fosters a multifaceted hydrological setting. A variety of geological formations and soil types within the region contribute to the mixed pattern of infiltration rates, groundwater recharge and surface runoff (Jehn et al., 2021) .

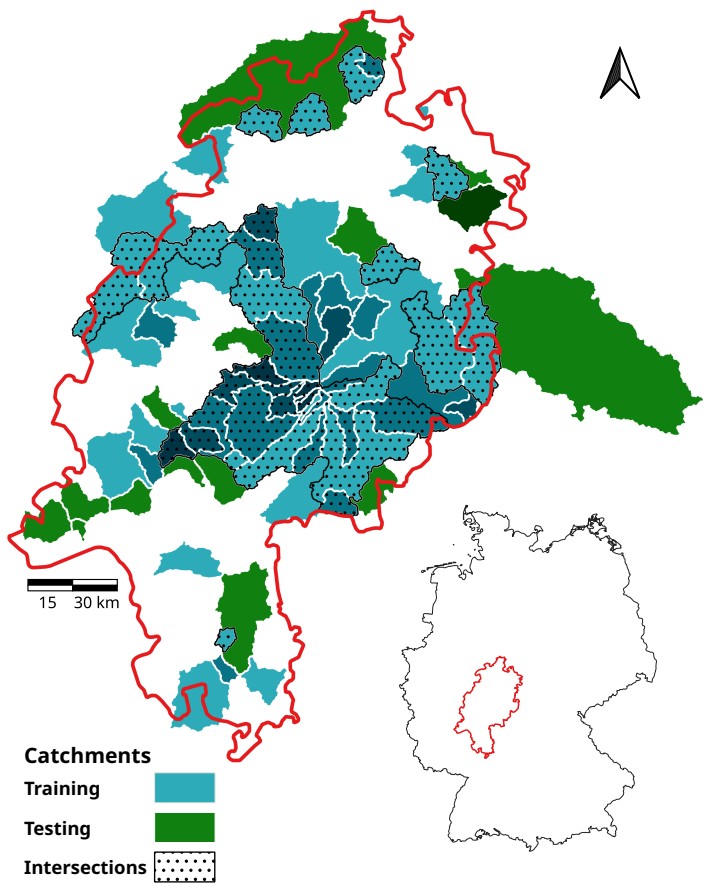

**Figure 1.** Geographic distribution of the catchments in Hesse and Hesse's location within Germany. Darker shades represent nested catchments, while intersections indicate catchments partially incorporated in both training and testing phases.





**Table 2.** Summary of Daily Forcing Data and Static Catchment Attributes Utilised for Modelling: Detailing the Spatial Resolution of the Original Data Sources with the Aggregation Methods and the Respective Units.

| Feature | Spatial resolution | Aggregation | Unit |
|---|---|---|---|
| precipitation | 1,000 m | daily sum | mm |
| evapotranspiration | 1,000 m | daily sum | mm |
| soil temperature (5 cm) | 1,000 m | daily mean | °C |
| soil type | 1:500,000 | spatial majority | classes (n=5) |
| soil texture | 1:1,000,000 | spatial majority | classes (n=4) |
| geology type | 1:250,000 | spatial majority | classes (n=2) |
| land use | 1:100,000 | spatial majority | classes (n=3) |
| permeability | 1:250,000 | spatial majority | classes (n=6) |
| average precipitation | 1,000 m | annual mean | mm |
| catchment size | 40 m | at reach pour point | $m^2$ |
| elongation ratio | 40 m | at reach pour point | / |
| soil depth | 1:1,000,000 | spatial mean | m |
| average slope | 40 m | spatial mean | ° |
| average evapotranspiration | 1,000 m | annual mean | mm |

## 2.2 Data Sources

The data set used in this study is derived from Jehn et al. (2021). For each catchment, daily sum of precipitation [mm], daily sum

of evapotranspiration [mm] and soil temperature in 5 cm soil depths [°C] are available along with the corresponding discharge [mm]. In addition, the data set includes 11 static catchment features corresponding to every catchment (Table 2). As suggested by Kratzert et al. (2019a), the inclusion of static catchment attributes can improve the performance of machine learning models. Table 2 provides an understanding of the underlying aggregation of data, spatial resolution and units. Apart from discharge data, which is accessible upon contacting the Hessian Agency for Nature Conservation, Environment and Geology, all other

data sets are publicly available within the associated repository of Jehn (2020).

## 2.3 Data preprocessing

The preprocessing of the input data is an essential step, as it ensures that the quality and integrity of the data is maintained. This process entails a detailed analysis of data continuity, encoding nonnumerical values, splitting the data set into training and validation subsets, followed by data normalisation and subsequent transformation. The data analysis revealed discontinuities

in the discharge data across the time series of 39 catchments. In order to provide the longest possible time series for the





training process, a total of 54 out of the full set of 95 catchments were selected for model training. These catchments cover 28 years (1991–2018). Of the remaining 39 catchments, 35 were utilised for testing, each with a temporal resolution spanning six years from 1997 to 2002. Rivers containing artificial constructions that impede discharge through impounds (e.g., reservoirs) were not considered in this analysis. However, it should be noted that a subset of the selected rivers might be

equipped with hydraulic control mechanisms, such as floodgates (Jehn et al., 2021). For both training and testing data sets, all categorical features (Table 2) were encoded with label encoding. For that, every unique variable of a categorical feature was replaced by a non-repeatable integer value (Lin et al., 2020). This approach was preferred over the frequently recommended one–hot–encoding technique (Duan, 2019; Cerda and Varoquaux, 2022) to circumvent an increase in the total feature count equivalent to the number of unique feature variables, as occurs with one–hot–encoding (Ul Haq et al., 2019). Moreover, label

encoding accommodates ordinal scales, which is better suited for hierarchical features such as permeability. However, Potdar et al. (2017) indicate that label encoding yields the lowest performance in the context of various investigated encoding methods. Consequently, it cannot be unequivocally asserted that this method stands as the optimal approach. The training data set of 54 catchments was then further divided, using 80% of the data for training and 20% for validation. Subsequently, the two data sets were normalised by employing the min–max scaling method, with a range of $[0, 1]$ chosen as the boundaries. Concurrently, the

precision of the data representation was configured to adhere to a float32 format. The target variable was scaled independently of the features. Moreover, to prevent data leakage, each feature normalisation was established solely based on the training data set. The normalised training data set exhibited a shape of $N \times D$ for each catchment, where $N$ signified the number of samples in time and $D$ represented the number of features. To assess the impact of additional static features, two distinct data sets were created. The first data set included only three features with daily forcing data and assumed a shape of $N \times 3$, while the

second one incorporated all 11 static features and took a shape of $N \times 14$. A two–dimensional moving window, characterised by dimensions $T \times D$, was subsequently implemented, where $T$ represents the moving window size, also known as look–back period or sequence length (Figure 2). This window is continuously incremented by a single period in the dimension of $N$, with the initial window encompassing observations $[N_1, N_T]$. The consecutive window encapsulates observations $[N_2, N_{T+1}]$, this pattern is maintained until the window reaches the final element of the data set ($N_n$). Consequently, the entire data set was

partitioned into $m = N_{n-T+1}$ subsamples for every catchment. All subsamples were combined into a three–dimensional array ($N_{n-T+1} \times T \times D$). The transformed catchment data sets were stacked to one final training set with the shape of $C \times N_{n-T+1} \times T \times D$, where $C$ was equal to the number of catchments. The identical transformation was implemented for both validation and test data sets, encompassing those with and those without static features. It is important to know that the transformation of the data is already part of the hyperparamterisation process, a concept further elucidated below.

## 2.4 Hyperparametrisation

The performance of machine learning models is remarkably influenced by the optimisation of their respective hyperparameters (Shekhar et al., 2022; Ozaki et al., 2021). In the domain of machine learning, hyperparameters are variables that define the configuration of the models and are set prior to the training process (Bhattacharjee et al., 2021), while the term parameter





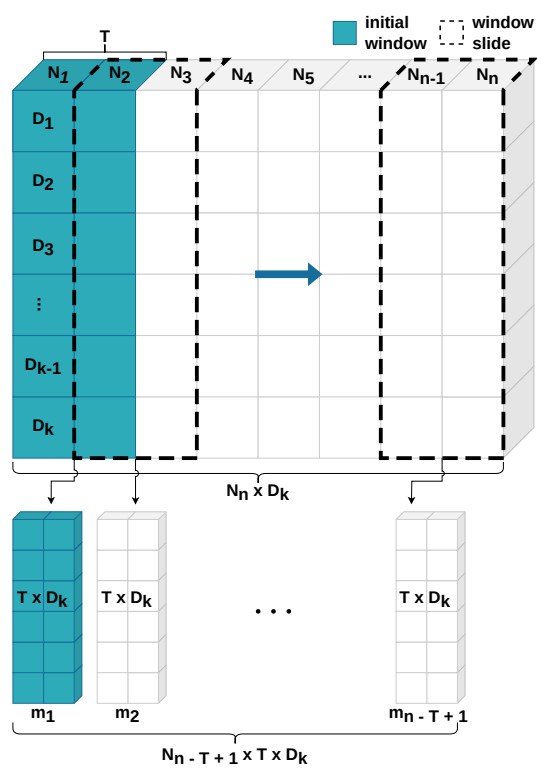

**Figure 2.** Schematic procedure of data transformation by applying a moving window: This procedure primarily involves the partitioning of the data into distinct sections, employing a window (blue) that slides across the data set, effectively creating a temporal snapshot ($m$). $T$ delineates the window size within the temporal dimension, $D$ represents the feature dimension, and $N$ signifies the temporal samples with a daily resolution.

refers to the variables that the model learns via training (Goodfellow et al., 2016). The selection of an appropriate tool for

hyperparameter optimisation is a critical step. Consequently, this task was conducted utilising a Python framework known as Spotpy (Houska et al., 2015). The framework offers computational optimisation techniques for calibrating models such as a Latin Hyper Cube Sampler (LHS), an appropriate method for selecting input variable values within a specified range, given its ability to generate near–random samples from a multidimensional hyperparameter distribution (McKay et al., 1979). The hyperparameters of the models are contingent upon the architectural design.

In this study, three distinct model architectures were explored: LSTM, GRU and CNN. LSTM and GRU are both types of Recurrent Neural Networks (RNNs) and are specifically designed to handle sequential data, such as time series. Because the employed LSTM and GRU models possess an identical layer structure with the exception of the recurrent layer, both models share an equivalent set of hyperparameters. A detailed overview of the utilised hyperparameters can be found in Table 3. The hyperparameter $T$ denotes the window size employed in the moving window mechanism and signifies the length of the





sequence, representing how many time steps (past days) are used to predict the discharge of the following day. This sequence encapsulates the historical information considered during prediction. The feature maps $F$ quantify the number of results or features generated within the convolution process. This is achieved by utilising a kernel of size $k$, referred to as the filter size, which is systematically applied over the data to extract essential patterns and characteristics, thereby transforming the input data. In the context of LSTM and GRU models, the unit $U$ refers to the number of hidden neurons within the RNN layer.

This quantity not only characterises the internal complexity of the layer but also corresponds to the output dimension. The last hyperparameter under consideration is the dropout rate $p$, which represents the fraction of the neurons that are randomly set to zero during training (Srivastava et al., 2014).

     The ranges of the hyperparameters were delineated in preliminary experiments by repeatedly training each model employing LHS over wider ranges. Any hyperparameter that fell below or exceeded the minimum and maximum bounds of Table 3

respectively, demonstrated inferior performance on average. The final training process was executed with a sampling size of 100 for each model and batch size combination, with and without static features. This culminated in a total of twelve distinct sampling processes.

**Table 3.** Ranges of hyperparameters deployed across different neural network models within the Latin Hypercube sampling framework.

| Model | Hyperparameter | Min | Max |
|---|---|---|---|
| CNN | Window size (T) | 50 | 300 |
| | Feature maps (F) | 100 | 500 |
| | kernel size (k) | 3 | 9 |
| LSTM / GRU | Window size (T) | 50 | 300 |
| | Units (U) | 10 | 500 |
| | Dropout rate (p) | 0.05 | 0.5 |

## 2.5    Model architectures

The architecture of the LSTM was first introduced by Hochreiter and Schmidhuber (1997). An LSTM consists of a memory

cell governed by four specific gate units, thus granting the capacity to preserve information over extended periods (Cho et al., 2014). Through this architectural design, LSTMs possess the capability to mitigate the challenges associated with exploding or vanishing gradients, as encountered with traditional RNNs. While the nuanced workings of LSTM cells and their concomitant advantages are pertinent, they have been extensively discussed in prior research and thus will not be repeated within this study (Hochreiter and Schmidhuber, 1997). The architectural design of a GRU model is inspired by the structure of LSTMs

with the distinction that it incorporates only two gates to regulate the information flow. This results in reduced computational complexity and thereby rendering GRU more computationally efficient while still addressing the exploding/vanishing gradient problem (Cho et al., 2014). In contrast, CNNs are tailored for grid–like data structures, including images. The CNN architecture





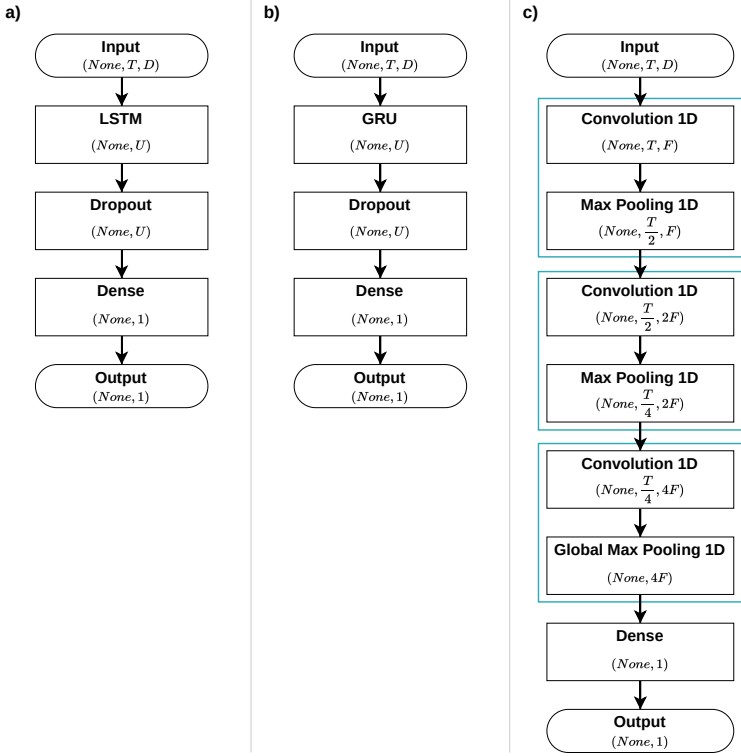

**Figure 3.** Schematic diagrams of the architectures of the three utilized models: (a) Long Short-Term Memory (LSTM), (b) Gated Recurrent Unit (GRU), and (c) Convolutional Neural Network (CNN).

was first introduced by Fukushima (1980). The term convolutional neural network was introduced by LeCun et al. (1989), who developed a model for handwritten digit recognition. CNN models possess a significant benefit in that the convolution operation

is inherently parallelizable, allowing for the simultaneous execution of numerous calculations. An additional merit is the ability to extract features, irrespective of the exact location where the feature was found. This reduces the number of input samples needed for training the network size and thus further improves computational efficiency (Lecun et al., 1998). Note that these extracted features are not the same as features listed in Table 2. The architectural configurations of the three models employed in this study are depicted in Figure 3, with further explanations provided in the subsequent sections.

**2.5.1  LSTM**

The LSTM model comprises a single LSTM layer configured with a designated number of hidden units ($U$). To mitigate overfitting and promote generalisation, a dropout layer is directly connected to the LSTM layer, introducing regularisation by randomly deactivating a specific fraction (dropout rate) of the hidden units (Srivastava et al., 2014). The final layer is a





dense layer that applies a Sigmoid activation function, which converts the output into a probability value between zero and
one (Figure 4c). This strategy was adopted to avert the potential of predicting negative discharge values. A comprehensive
examination of all activation functions employed within the models is provided in Figure 4. This illustration delineates the
specific characteristics of each function, highlighting that both the Rectified Linear Unit (ReLU) and Sigmoid functions are
designed to avoid negative values. The ReLU function, in particular, suppresses negative values by setting them to zero, while
the Sigmoid function, recognised by its characteristic S–shape, maps any input into values between zero and one. Pertinent
to the context of deep learning, especially image recognition, ReLu is often favoured for its expedited learning capabilities,
yielding enhanced performance and superior generalisation attributes (Krizhevsky et al., 2017). However, it has been observed
in preliminary experimental setups that the Sigmoid function exhibits a greater degree of stability, while ReLU demonstrated
a higher propensity to induce gradient exploding. The complete architectural design of the LSTM model is illustrated in
Figure 3a.

### 2.5.2 GRU


The architecture of the GRU model shares a structure similar to that of the previously described LSTM model, with the primary
difference being the substitution of the LSTM layer with a GRU layer (Figure 3b). Similar to the LSTM model, the GRU model
contains a single layer configured with a designated number of hidden units ($U$) and employs a dropout layer directly connected
to the GRU layer to mitigate overfitting and promote generalisation. Finally, this model also incorporates a dense layer with a
Sigmoid activation function.

### 2.5.3 CNN

The CNN is composed of a series of three convolution cells, each containing a one–dimensional convolution layer followed
by a pooling layer. The convolution layers incorporate a ReLU activation function (Figure 4b) and employ a sliding window
mechanism known as a kernel that traverses the input data for processing. As previously elucidated, this kernel is responsible
for extracting feature maps ($F$) from time–dependent input features. The kernel, with a size of $k$, is applied uniformly across
all convolution layers. In each successive convolution layer, the quantity of feature maps is increased by a factor of two,
thereby increasing the model's capacity to extract and represent complex features. In the initial pair of convolution cells, the
temporal dimension ($T$–array) within the pooling layer is reduced by a factor of two by employing a stride of size two across
each $T$–array, while the third pooling layer extracts a single set of feature maps along the temporal axis of all $T$–arrays. To
preserve the temporal dimension during the convolution process, each convolution layer incorporates symmetric zero–padding.
This technique involves adding zeros around the input data, ensuring that the processed dimension remains unchanged after
applying the convolution operation. The last layer of the model is a dense layer that compresses the model dimensions to
produce a single output value for each prediction. This layer is fully connected to the preceding layer and uses a leaky rectified
linear unit (LeakyReLU) activation function as depicted in Figure 4b. The LeakyReLU, akin to the standard ReLU (shown in





the same figure), distinguishes itself by incorporating a small slope for negative values. This characteristic enhances gradient propagation and mitigates the issue of vanishing gradients (Ramachandran et al., 2021). The selection of the LeakyReLU over the standard linear activation function (Figure 4a) was driven by the latter's propensity to generate negative predictions for the discharge values. Although LeakyReLU does not entirely preclude negative predictions, it effectively modulates them into marginally negative outputs and therefore reduces the extent of negative predictions. Although the Sigmoid function is

effectively utilised in LSTM and GRU models to prevent negative discharge predictions, its application within the CNN model framework yielded suboptimal results in preliminary trials, especially when compared to the performance achieved using the LeakyReLU activation function. This informed the decision to opt for LeakyReLU in our work. A visual representation of the complete architectural design of the CNN model is presented in Figure 3c.

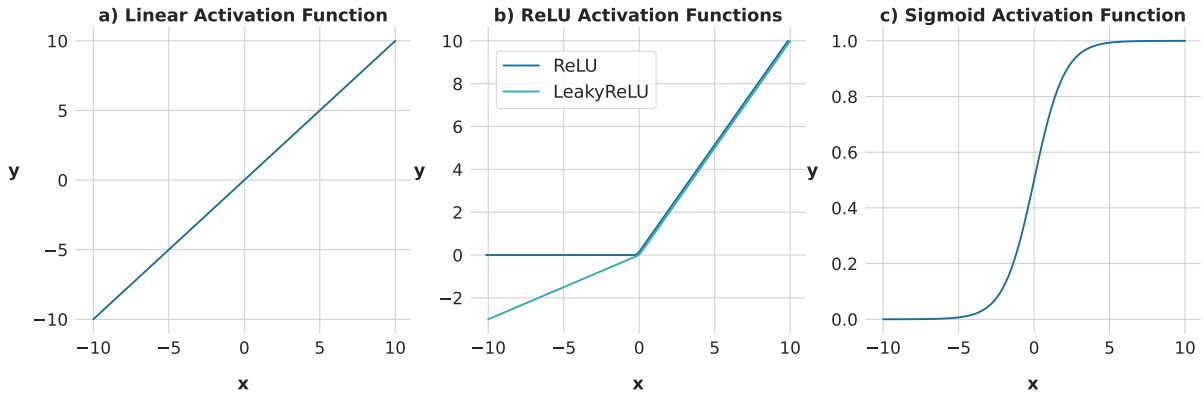

**Figure 4.** Visualization of the three activation functions utilized within the employed models. The diagrams show the graphical representations and functional ranges of (a) the linear function, which preserves the raw, untransformed input; (b) the Rectified Linear Unit (ReLU) function, which maps negative inputs to zero and passes positive inputs unchanged; and (c) the sigmoid function, characterized by its distinct 'S'-shape, which compresses any input into a range between zero and one. Note: different Y-axis scales.

## 2.6 Loss function

In machine learning algorithms, the role of the loss function is paramount as it quantifies the discrepancy between the model's predictions and the actual data (Wang et al., 2022). The loss function is regulated by an algorithm known as the optimizer. This optimizer strives to enhance the model performance by iteratively determining the loss and then adjusting the model parameter to reduce this loss. This is achieved by identifying the gradient or derivative of the loss function, which denotes the local minimum (least steep ascent). Thus, by minimising the loss, the machine learning model can improve its predictive

accuracy and thereby enhance its capacity to generalise from the training data to unseen instances. The optimizer used for all utilised models in this study is the Adam–optimizer. This algorithm provides high computational efficiency for gradient–based optimisation and is suitable for large models that include a high number of parameter sets (Kingma and Ba, 2017). The choice





of loss function is dictated by the specific task at hand. A commonly used loss function when predicting continuous data is the Mean Square Error (MSE), which is favoured for its computational efficiency. However, MSE suffers from sensitivity to outliers due to its quadratic penalty and exhibits scale–dependence, rendering it less interpretable and comparably challenging when evaluating models across disparate output scales (Liano, 1996; Gupta et al., 2009). Another metric used to capture model performance, traditionally used in hydrology, is the Nash–Sutcliffe efficiency (NSE) (Knoben et al., 2019). Based on the close similarities between MSE and NSE and hence the inherent disadvantages, NSE is not an ideal choice as loss function either (Gupta et al., 2009). To mitigate the systematic issues encountered in optimisation processes that arise from formulations linked to the MSE or NSE, we decided to utilise the more resilient Kling–Gupta efficiency (KGE). The KGE corrects for underestimation of variability, by providing a direct evaluation of four different facets of the discharge time series, which encompass shape, timing, water balance and variability (Santos et al., 2018). The definition of KGE is delineated in Equation 1.

$$KGE = 1 - \sqrt{(r-1)^2 + (\alpha - 1)^2 + (\beta - 1)^2}$$  (1)

with:

$$r = \frac{Cov_{(\text{obs, sim})}}{\sigma_{\text{obs}} \cdot \sigma_{\text{sim}}}$$

$$\alpha = \frac{\sigma_{\text{sim}}}{\sigma_{\text{obs}}}$$

$$\beta = \frac{\mu_{\text{sim}}}{\mu_{\text{obs}}}$$

where $\mu$ is the mean, $\sigma$ is standard deviation, and $r$ is the linear correlation factor between observations and simulations. The variable $\alpha$ is a measure of how well the model captures the variability of the observed data and $\beta$ defines a bias term indicating how much the model's predictions systematically deviate from the true values (Knoben et al., 2019). Analogous to NSE, KGE also indicates the highest performance when equal to one. However, the goal of the loss function is to minimise the error; thus, the discrepancy between simulation and observation should approach zero. Therefore, the implemented loss function $L$ results in Equation 2.

$$L_{(\text{obs, sim})} = \sqrt{(r-1)^2 + \left(\frac{\sigma_{\text{sim}}}{\sigma_{\text{obs}}} - 1\right)^2 + \left(\frac{\mu_{\text{sim}}}{\mu_{\text{obs}}} - 1\right)^2}$$  (2)

## 2.7 Model training

The training process was conducted using a GeForce RTX 3090 graphics card equipped with 24 GB of memory. Each model was subjected to training with batch sizes of 256 and 2,048. The batch size is a fraction of the total number of training samples and represents the number of samples utilised to train the model prior to an update of the internal parameters (Radiuk, 2017). The batch size has no physical interpretation in the context of hydrological processes but functions as a crucial hyperparameter





in the training of neural networks. Prior studies, such as Kratzert et al. (2019a, b), have demonstrated the successful application of a batch size of 256. In this study, this batch size was also adopted and served as the baseline. To further explore the impact of larger batch sizes, a multiples of 256 was employed. A batch size of 2048 was then utilized, as this represents the upper limit of the memory capacity of the graphics card used.

The maximum number of epochs designated for training was set to 60. An epoch refers to a single iteration over the entire 260 training data set during which the model's parameters are adjusted to minimise loss. However, the training process was configured to terminate when the validation loss failed to show improvement throughout five consecutive epochs. An enhancement was recognised when the validation loss decreased by a minimum of 0.001 during these five epochs. This mechanism is called early–stopping. Given that the input data for the training procedure are arranged by catchments, shuffling of data was implemented to circumvent the potential for overfitting to a specific catchment. Furthermore, each model was trained both with and 265 without the inclusion of static features for the two specified batch sizes. This leads to a total of four distinct training phases for every model with a specific hyperparameter set. The static features were analogously processed within the models to the treatment of the daily features. The learning rate, frequently acknowledged as the paramount hyperparameter to tune, exerts a considerable influence on the training of models that employ gradient descent algorithms (Xu et al., 2019). Hence, when the learning rate is too high, the optimizer may diverge from the local minimum, while setting it too low can result in a protracted 270 learning process (Zeiler, 2012). To efficiently address this behaviour, a dynamic adjustment of the learning rate was integrated into the training process using a learning rate scheduler. This algorithm modifies the learning rate based on the current epoch number. During the warm–up period, the learning rate linearly increased from the initial–rate to the base–rate throughout three epochs. The warm–up period is followed by a decay period lasting ten epochs, during which the learning rate linearly decreases from the base–rate to the minimum–rate. Following the decay phase, the learning rate is kept constant at the minimum–rate for 275 the remaining epochs. Detailed information can be found in Table 4.

**Table 4.** Gradual alterations in the learning rate throughout the 60 epochs of the model training process.

| Epoch | Stage | Learning Rate |
|-------|-------|---------------|
| 1-3 | Warm up | Linear increase from $1e^{-6}$ to $5e^{-4}$ |
| 4-13 | Decay | Linear decrease from $5e^{-4}$ to $5e^{-5}$ |
| 14-60 | Cool down | Constant $5e^{-5}$ |





## 3  Results and Discussion

### 3.1  Model Performance

The analysis depicted in Figure 5 delineates a comparative evaluation of model efficiency concerning architectural variations, batch sizing, and the incorporation of supplementary static attributes. The findings reveal that employing CNN models in con-

280 junction with static features yielded a mean KGE of 0.80 and 0.78 for batch sizes of 256 and 2,048, respectively. The inclusion of static features provides a performance benefit because the mean accuracy drops to 0.71 and 0.67 when static features are omitted for batch sizes of 256 and 2,048, respectively. This aligns with the findings presented by Kratzert et al. (2019b), which assert that static catchment attributes contribute to enhancing the overall model performance. Notably, the maximum KGE in the absence of static features reached 0.97 and 0.92 for batch sizes of 256 and 2,048, respectively, highlighting the potential for

high model performances even without static features. On the contrary, the minimum KGE drops when omitting static features to -0.21 and -0.26 for batch sizes of 256 and 2,048, respectively, showing the lowest minimum performance of all models. This suggests a deficiency in the model's ability to generalise, a phenomenon frequently observed when overfitting occurs (Srivastava et al., 2014). Regarding the minimum KGE values, when utilising static features, the CNN models demonstrated the third and fourth highest minimum values, registering at 0.24 and 0.20 for batch sizes of 256 and 2,048, respectively.

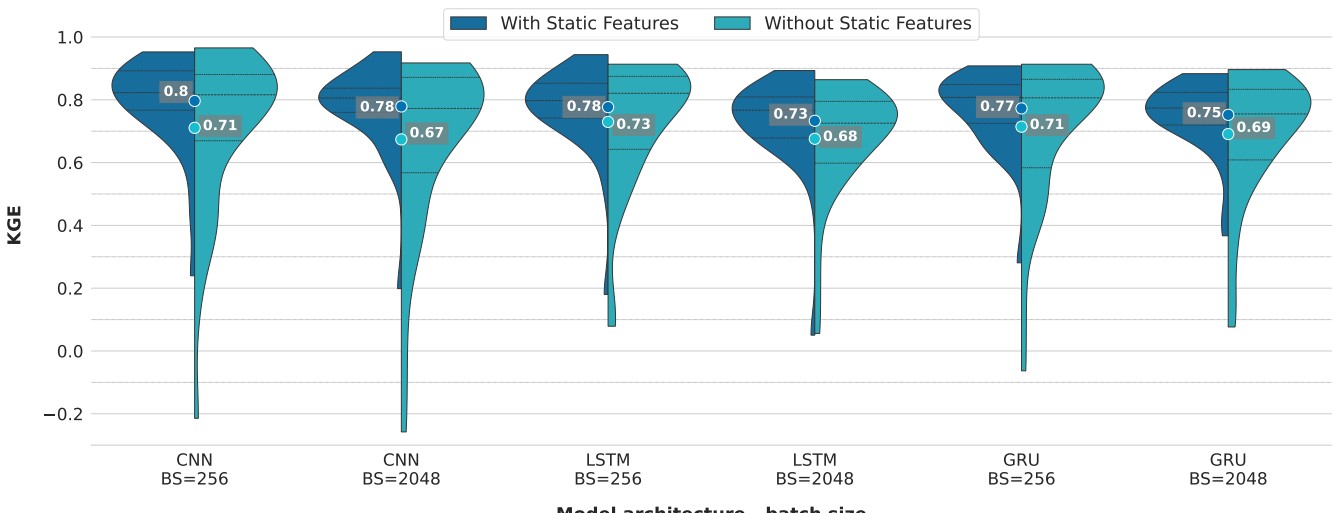

**Figure 5.** Evaluation of performance discrepancies in the applied models relative to batch size and additional static catchment attributes during the testing period. The number represents the average KGE over all 35 catchments. The dotted line displays the percentile intervals.

In the case of LSTM networks, mean KGE values of 0.78 and 0.73 with static features for batch sizes of 256 and 2,048, respectively, can be noted. The mean KGE declined to 0.73 and 0.68 when static features were omitted for batch sizes of 256 and 2,048, respectively. Notable is the maximum performance achieved with static features, which reached 0.94 for a batch





sizes of 256. In contrast, the LSTM with a batch size of 2,048 exhibited the lowest minimum value of 0.05 across all models with static features. For models run without static features, the LSTM with a batch size of 256 recorded the highest minimum value of 0.09. Conversely, the LSTM model with no static features and a batch size of 2,048 presented the lowest maximum KGE of 0.86.

For GRU, the mean KGE exhibited similar trends with the inclusion of static features, reaching 0.77 and 0.75 for batch sizes of 256 and 2,048, respectively. The mean performance declined to 0.71 and 0.69 when static features were omitted for batch sizes of 256 and 2,048, respectively. The GRU model with a batch size of 2,048 demonstrated the highest minimum KGE value of 0.37 among all models when static features were incorporated. Following closely, the GRU model with a batch size of 256 under the same feature scenario presented the second–highest minimum KGE of 0.28. Upon examining the performance range, the GRU model with static features and a batch size of 2,048 exhibited the narrowest performance range of 0.52. Subsequently, the GRU model with static features and a batch size of 256 displayed a performance range of 0.63, indicating robust generalisation capabilities for these two models. Notably, for both batch sizes, the GRU model demonstrated a marginally higher maximum KGE when static features were omitted. This finding contradicts the outcomes of all other models, where the inclusion of static features consistently reduced the maximum KGE, regardless of the batch size. The sole exception to this pattern was observed in the CNN model with a batch size of 256 utilising no static features.

All together, when analysing the influence of batch size across various models, it becomes evident that an increase in batch size correlates with a decrease in performance. This observation is confirmed by the study of Masters and Luschi (2018), who discovered that smaller batch sizes contribute to enhanced training stability and generalisation performance when employing CNN models for image classification. Additionally, Kandel and Castelli (2020) identified a strong correlation between learning rate and batch size, proposing that higher learning rates should be employed when utilising larger batch sizes. However, the learning rate remained constant across varying batch sizes throughout this study.

Altogether, these results suggest that:

i.  Smaller batch sizes contributed to better model performance with regard to mean KGE values.

ii. Static features generally improved the mean KGE across all architectures and batch sizes.

iii. The CNN model with static features and a batch size of 256 showed the highest mean KGE and therefore slightly outperforms LSTM and GRU models.

iv. The KGE performance ranges for models with static feature are substantially smaller and on a higher level than the ranges for models without static features.

v.  Overall, the GRU model with a batch size of 256 and static features exhibited favourable KGE performances akin to LSTM and CNN models and mitigated poor predictions across all test catchments.





**Comparing evaluation metrics**

To further investigate the efficacy of the applied models, additional performance metrics were incorporated. Among these, the
NSE was selected to facilitate comparison with prior studies that conventionally utilise this metric. Moreover, the Percent Bias
(PBIAS) was employed to gauge the systematic deviation of the modelled data from observed values, indicating whether the
model predictions consistently overestimate or underestimate the observations (Moriasi et al., 2007). The Mean Absolute Error
(MAE) was integrated as a metric to quantify the absolute discrepancies between model predictions and actual observations,
serving as a direct assessment of model precision (Siqueira et al., 2016). Lastly, the Coefficient of Determination ($R^2$) was
adopted as an indicator for evaluating the degree of alignment between simulations and observed data, reflecting the model's
'goodness–of–fit' (Onyutha, 2022). A comparative view of the results of all the used performance metrics is shown in Table 5.
Overall, the presented data indicates, that NSE metrics are marginally lower than the KGE values. This phenomenon could
potentially stem from the presence of counterbalancing errors, an inherent limitation associated with KGE metric. Such coun-
terbalancing errors materialise through concurrent overestimation and underestimation of the predicting target. Given that bias
and variability collectively constitute two-thirds of the KGE, their effects may augment the aggregate score, without necessarily
indicating a more accurate or relevant model (Cinkus et al., 2022).

Notably, the CNN and LSTM models, when configured with a batch size of 256 and incorporating static features, achieved
the highest NSE values of 0.76 and 0.75, respectively. In comparison, the GRU model under identical configurations exhibits a
slightly inferior performance, marked by an NSE of 0.72. In the context of existing literature, a study by Nguyen et al. (2023a)
reported an NSE of 0.66 for an LSTM model calibrated across three distinct catchments, albeit with separate calibrations
for each and not extending to ungauged scenarios. Kratzert et al. (2019a) documented an NSE of 0.54 for an LSTM model,
which, despite being lower, is deemed more robust due to its validation across 531 catchments using k-fold cross-validation.
Nonetheless, the observation that NSE values surpassing 0.7 in the most efficacious model for each architecture serves as a
testament to the proficiency of these artificial models, contingent upon adept hyperparameter optimisation and the availability
of ample data for learning processes.

All CNN models universally exhibit a positive PBIAS, signifying a consistent underestimation of discharge rates, regardless
of variations in batch size or feature scenarios. Notably, CNN models lacking static features manifest on average smaller
discharge of approximately 7%, marking them as the models with the most significant underestimations. Conversely, the CNN
model employing a batch size of 256 alongside static features demonstrates the smallest PBIAS, recorded at 0.06%.

In contrast, LSTM models display a PBIAS pattern that does not adhere to a discernible trend. The LSTM model achieving
the highest KGE metric overestimates the discharge by an average of 3.46%. The LSTM models with a batch size of 2,048 and
inclusion of static features exhibits the most substantial overestimation, with a PBIAS of -5.1%. The absence of static features
in LSTM models tends to yield PBIAS values closer to zero, which is preferable.

GRU models reveal a negative PBIAS when static features are incorporated and positive PBIAS without them. The most
favourable PBIAS among GRU models, -0.48%, is observed in the model with a batch size of 256 and static features, closely





**Table 5.** Synthesis of performance metrics across models, batch sizes, and feature scenarios during the testing period. Numbers shaded blue denote higher scores for each metric.

| Model | Batch size | Features | KGE | NSE | PBIAS | MAE | $R^2$ |
|---|---|---|---|---|---|---|---|
| CNN | 256 | +SF | 0.8 | 0.76 | 3.82 | 0.29 | 0.82 |
| | 2048 | | 0.78 | 0.72 | 0.06 | 0.3 | 0.79 |
| | 256 | -SF | 0.71 | 0.66 | 7.13 | 0.3 | 0.84 |
| | 2048 | | 0.67 | 0.61 | 7.78 | 0.32 | 0.82 |
| LSTM | 256 | +SF | 0.78 | 0.75 | -3.46 | 0.3 | 0.82 |
| | 2048 | | 0.73 | 0.63 | -5.1 | 0.4 | 0.71 |
| | 256 | -SF | 0.73 | 0.7 | 1.87 | 0.31 | 0.82 |
| | 2048 | | 0.68 | 0.59 | -1.21 | 0.39 | 0.72 |
| GRU | 256 | +SF | 0.77 | 0.72 | -0.48 | 0.32 | 0.79 |
| | 2048 | | 0.75 | 0.69 | -2.75 | 0.37 | 0.77 |
| | 256 | -SF | 0.71 | 0.67 | 1.65 | 0.32 | 0.82 |
| | 2048 | | 0.69 | 0.59 | 1.24 | 0.39 | 0.73 |

aligning with the best–performing CNN model's PBIAS of 0.06%. Overall, GRU models display the least average deviation in PBIAS.

Regarding MAE, most models exhibit comparable outcomes with an MAE around 0.3 mm. However, LSTM and GRU models with a batch size of 2,048 are exceptions, showing a slightly elevated MAE around 0.4 mm. Despite this, the models generally demonstrate an ability to minimise this error metric, particularly evident in CNN models with higher PBIAS values where the cancellation of positive and negative predictive errors does not occur.

The $R^2$ scores of every model architecture show always a slightly better fit without static features, when comparing equal batch sizes. One exception to this trend are the GRU models with a batch size of 2048, where the model incorporating static features shows a higher fit than without static features. Furthermore, the $R^2$ values confirm the analysis of the KGE performance, which showed better performance with smaller batch sizes.

Summarising the insights of Table 5 corroborates that CNN models, when incorporating static features, manifest superior efficacy, particularly in the context of the metrics assessed for validation.

## 3.2 Runtime

To investigate the computational efficiency associated with the models employed, the runtime was measured for each model, considering variations in both batch size and the combination of features.



Both the batch size and the integration of additional static features significantly influence the runtime of models across all employed architectures, as evidenced in Figure 6. The CNN model with a batch size of 2,048 and without static features presented the shortest runtime of approximately 2.3 minutes. Although the CNN model demonstrated rapid convergence towards its optimal minimum error, it simultaneously exhibited the lowest performance as delineated in Figure 5. This suggests that the

375 conditions were not sufficiently robust to discern the intrinsic patterns. Using an identical batch size and feature configuration, the GRU model, along with the CNN model configured with a batch size of 256 and no static features, had the second shortest runtimes of approximately 4.2 minutes.

The introduction of static features resulted in a notable increase in the runtime for all models, barring the GRU model with a batch size of 256, where the inclusion of static features marginally reduced the runtime, rendering it the fastest among all

380 models that utilised static features. The runtime augmentation was especially pronounced in the CNN model with a batch size of 2,048, showing a more than twelve fold increase, thereby marking it as the most time–consuming model across all evaluated scenarios. LSTM models exhibited also a substantial increase in runtime across both batch sizes upon the incorporation of static features.

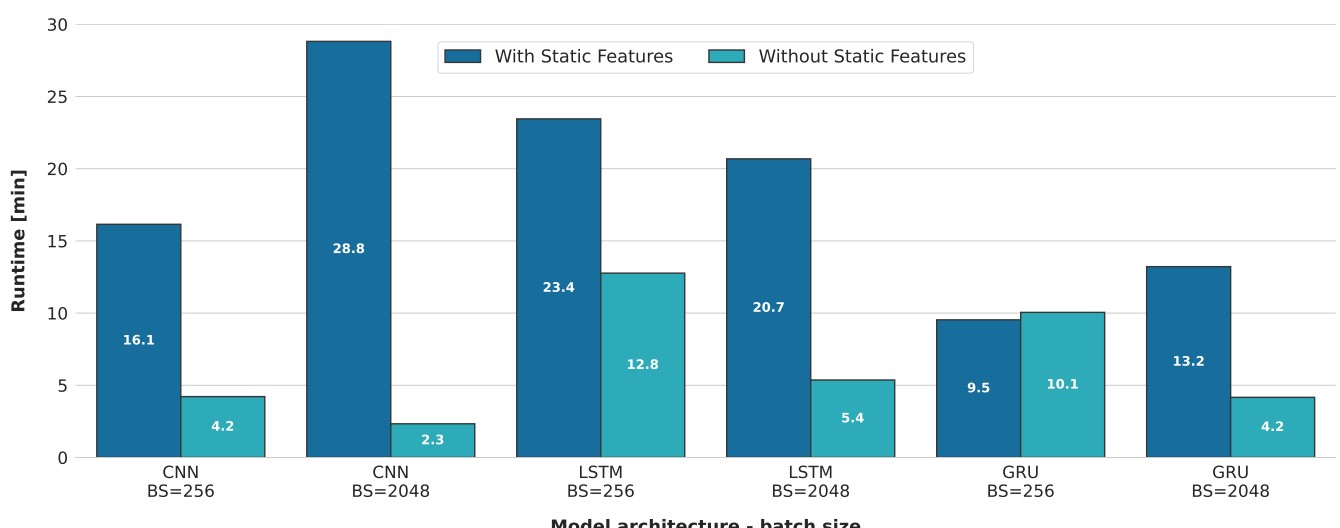

**Figure 6.** Comparison of model runtime across three different architectures (CNN, LSTM, and GRU) with varying batch sizes (256 and 2048) and the presence or absence of static features.

Within identical model architectures, it is observed that larger batch sizes contribute to faster runtimes in the absence of

385 static features. Conversely, when static features are employed, models tend to exhibit faster runtimes with smaller batch sizes, with the exception of the LSTM models. For these models, an escalation in batch size consistently results in accelerated runtimes, irrespective of the feature configuration. The different behaviour of additional features towards training runtime while using different batch sizes is unexpected and cannot be explained solely by considering the batch size and feature scenarios.



As reported by Radiuk (2017), larger batch sizes correlate with increased runtimes, which is attributable to the higher com-
putational utilisation required to process an increased quantity of training samples for the purpose of updating model weights.
Nonetheless, this assertion assumes that the models under comparison diverge only in terms of batch and feature size. This
presumption does not apply to the present study, where each model is also characterised by a unique optimised combination
of hyperparameters (Table 3). A possible explanation might be that all models exhibiting a more protracted runtime require
additional epochs to converge. This phenomenon could be facilitated by the early–stopping mechanism deployed in model
training, which permits the termination of the training process when the optimised metric ceases to demonstrate improvement.

Altogether, when static features are incorporated, the GRU model utilising a batch size of 256 demonstrates the fastest
runtime (9.5 minutes). In contrast, the CNN model, configured identically with respect to batch size and employed features,
exhibited a runtime of 16.1 minutes, consequently rendering the runtime of the GRU model 41% faster. In the final analysis, it
becomes evident that the GRU model exhibits superior runtime performance compared to both the CNN and LSTM models,
specifically when employing a batch size of 256 and utilising static features. In the context of RNN models, with a focus on
runtime, GRU models were found to be superior in efficiency compared to LSTM models. This stands in alignment to the
findings of Yang et al. (2020), who reported that GRU was 29% faster than LSTM when processing the identical data set.
However, as stated before, the examined models in this study exhibit disparities not only in terms of batch size but also encom-
pass other architectural parameters such as the number of utilised epochs, hidden units and the window size (Table 6). These
differences may result in altered computational efforts. Apart from the different model architectures, the specific configuration
of hyperparameters in each model yields varying computational effort. For example, an increase in window size results in a
more extended sequence to process, thereby necessitating additional computational effort. In the context of the CNN models,
the computational effort is contingent on the window size, feature maps, kernel size and the quantity of input features. Models
incorporating static features (+SF) possess 14 input features, whereas those without static features (-SF) contain only three
dynamic features. In contrast, the computational effort of the LSTM and GRU models is determined by the units within the
corresponding cell, the input feature size and the window size.

The observed increase in computational time for the GRU model, when running with a batch size of 256 and no static
features, is mainly due to a significantly larger window size, which increased from 87 to 298. This expansion, in the absence
of static features, requires a more extensive computational effort. In contrast, for CNN models employing a batch size of
2,048, the pronounced augmentation in execution time is primarily induced by an increase in the quantity of feature maps,
presenting a 2.3 fold increase. Generally, the marked prolongation in computational duration for CNN models incorporating
static features, as opposed to those excluding them, can be elucidated by the incorporation of a considerably higher number of
feature maps in the former. This enlargement is a direct consequence of the increased data volume processed by the models
when supplemented with static features. Notably, CNN models utilising a batch size of 2,048 manifest a reduction in window
size, implying that the model may encounter challenges in generalising from extended input sequences due to potentially
excessive variability among the samples within a batch. For the LSTM models with a batch size of 2,048, an 83% increase in
the number of hidden units, when static features are introduced, is the primary factor contributing to the substantial increase in





**Table 6.** Selection of utilized hyperparameters for the employed CNN, LSTM, and GRU models: A comparative examination of different feature scenarios, including scenarios with static features (+SF) and without static features (-SF), across two distinct batch sizes (256 and 2048).

| Model | Hyperparameter | Batch size 256 | | Batch size 2048 | |
|---|---|---|---|---|---|
| | | +SF | -SF | +SF | -SF |
| CNN | Window size (T) | 179 | 183 | 86 | 70 |
| | Feature maps (F) | 346 | 105 | 466 | 205 |
| | Kernel size (ks) | 4 | 6 | 8 | 8 |
| LSTM | Window size (T) | 232 | 288 | 168 | 159 |
| | Units (U) | 491 | 377 | 453 | 248 |
| | Dropout rate (p) | 0.37 | 0.34 | 0.29 | 0.23 |
| GRU | Window size (T) | 87 | 209 | 150 | 229 |
| | Units (U) | 373 | 364 | 480 | 172 |
| | Dropout rate (p) | 0.48 | 0.11 | 0.27 | 0.17 |

runtime for this configuration. Notably, the GRU model with a batch size of 256 and static features, which exhibits the smallest window size of 87 among all recurrent models, achieves the fastest runtime for models incorporating static features, a result directly attributable to its reduced window size, while still maintaining commendable predictive performance.

The architectural differences between CNN models and recurrent models (LSTM and GRU) render direct comparisons of their hyperparameter configurations impracticable, with the exception of window size. As indicated in Table 6, the window sizes of CNN models are smaller than those observed in recurrent models, except for the GRU model utilising a batch size of 256 and incorporating static features.

Moreover, an assessment of the best–performing models within each architecture (all configured with a batch size of 256 and incorporating static features) with regard to their hyperparameter configurations, reveals that it is the aforementioned GRU model that possesses the smallest window size (87), succeeded by the CNN (179) and LSTM (232) models. The increased length of input sequences implies greater computational demands, which partly accounts for the elevated runtime observed in the specified CNN model, despite its inherent capacity for parallel processing. As outlined in subsection 2.5, this attribute is typical of CNN models, in contrast to the sequential processing nature of LSTM and GRU models limits such parallelization.

In conclusion, the comparative analysis suggests that the GRU model, particularly with a batch size of 256 and the inclusion of static features, emerges as the optimal choice for hydrological applications that prioritise computational efficiency alongside predictive performance. Furthermore, the differential impact of batch sizes and feature configurations on the runtime across CNN, GRU, and LSTM models underscores the critical role of tailored hyperparameter optimisation in achieving computational efficiency without compromising model performance.





Given the observed favourable outcomes when utilising a batch size of 256 with static features, subsequent analyses will focus exclusively on models adhering to this configuration.

**Assessment of Flow Segment Performance**

To reinforce the analysis of performance, the recorded discharge data from all evaluated catchments, corresponding to the
highest–performing model within each architectural category, were divided into quartiles. First, the discharge data for each catchment were sorted in ascending order. Then, the sorted data were divided into four quartiles, with each quartile representing a 25% portion of the data range for each catchment, thereby forming four distinct segments. Subsequently, for each segment, KGE and PBIAS of the predicted discharge were calculated in relation to the observed values, as illustrated in Figure 7. Across all models, a noticeable increase in KGE is observed from the lowest to the highest flow segments, with the exception of Q2,
which represents lower flow levels and records the lowest KGE values. Remarkably, only within the highest flows is a positive KGE observed. This implies that the models predominantly discern peak flow events as critical data for learning, treating low flows as less significant or noise, which the models aim to diminish. This phenomenon may be attributed to a bias in the KGE towards elevated flows, thereby inadequately penalising inaccuracies in lower flow predictions. Consequently, forthcoming research should explore evaluation metrics that facilitate a more holistic optimisation approach. With regard to the highest
flows, the KGE metrics exhibit close resemblance across models, with the CNN model slightly leading with a KGE of 0.69. Conversely, the LSTM model demonstrates superior efficacy in modelling Q1 and Q2 flow segments.

Addressing the PBIAS, the pattern of enhanced model performance with increasing flow magnitudes, as noted with KGE metrics, persists. This is evidenced by the narrowing spread of the violin plots. Intriguingly, except for the Q4 segment, the PBIAS remains positive across all models for each flow segment, indicating a general underestimation of lowest to higher
flows and a mild overestimation of peak flows. Notably, the predictions by the CNN model for lowest flow exhibit the most pronounced bias, particularly on the positive spectrum, pointing to a lack of adequate generalisation capabilities.

A further decomposition of the KGE is illustrated in Figure 8, where each of the three components of the KGE (Pearson correlation coefficient (r), variability ($\alpha$), and bias ($\beta$)) are presented separately. These components offer insights into distinct aspects of the model's performance. The Pearson correlation coefficient (r) measures the strength and direction of the linear
relationship between the observed and simulated data. A value of 1 indicates perfect positive correlation, -1 indicates perfect negative correlation, and 0 indicates no correlation. The variability ($\alpha$) measures the ability of the model to capture the observed variability. A value of 1 indicates that the model's variability matches the observed variability. Values greater than 1 indicate the model has higher variability, while values less than 1 indicate lower variability. The bias term ($\beta$) indicates the systematic overestimation or underestimation by the model. A bias value of 1 means there is no bias, values greater than 1 indicate
overestimation, and values less than 1 indicate underestimation. Figure 8 reveals that r is more consistent across Q1 to Q4 for the LSTM model, unlike the CNN and GRU models, which display a wider range for r below 0.25. This indicates that the LSTM model is better at matching the timing of prediction for low flows. A similar trend is observed for $\alpha$, where the LSTM and GRU model exhibit higher variability, particularly for the lowest flows (Q1). However, the GRU model shows difficulties in



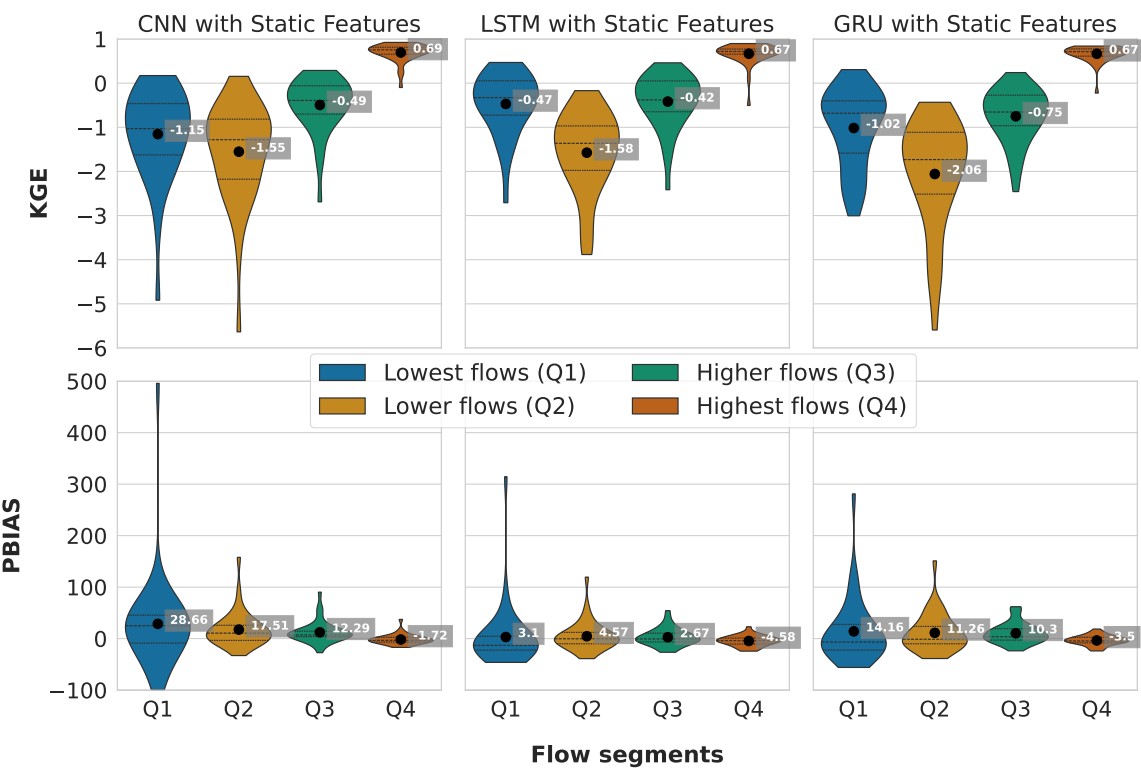

**Figure 7.** Comparative performance of CNN, LSTM, and GRU models incorporating static features across different flow segments. The top row displays the Kling-Gupta Efficiency (KGE) and the bottom row shows the Percent Bias (PBIAS) for the lowest flows (Q1), lower flows (Q2), higher flows (Q3), and highest flows (Q4). Each violin plot represents the distribution of model performance metrics for all evaluated catchments within each flow segment. The black dots indicate the mean values for each segment.

capturing variability for lower and higher flows (Q2 and Q3), with values of 3.96 and 2.63, respectively, compared to the LSTM

and CNN models. The bias term $\beta$ shows that the CNN model achieves the best score for the highest flows (Q4). Nevertheless, it also exhibits the largest bias for the lowest flows (Q1) among all models. Conversely, the LSTM model demonstrates superior performance for Q1 through Q3. Overall, this analysis suggests that the LSTM model exhibits favorable results across all KGE components. Appendix A presents the three best-performing and three worst-performing hydrographs of each model. Within the poorly performing hydrographs, it becomes evident that while the timing of the flow events is mostly accurate, the

magnitude is poorly captured, and the base flow is often underestimated. This suggests that these catchments might exhibit different hydrological behaviors compared to the better-predicted catchments, indicating the need for more diverse catchments in the training dataset.

In summary, the evaluation of flow segment performance has provided valuable insights into the performance distribution. While the CNN model showed superior average performance, as demonstrated within the preceding sections, the LSTM model





exhibited a higher degree of consistent performance across all flow segments. Additionally, the recurrent models displayed enhanced generalisation capabilities for the lowest flow rates in each catchment.

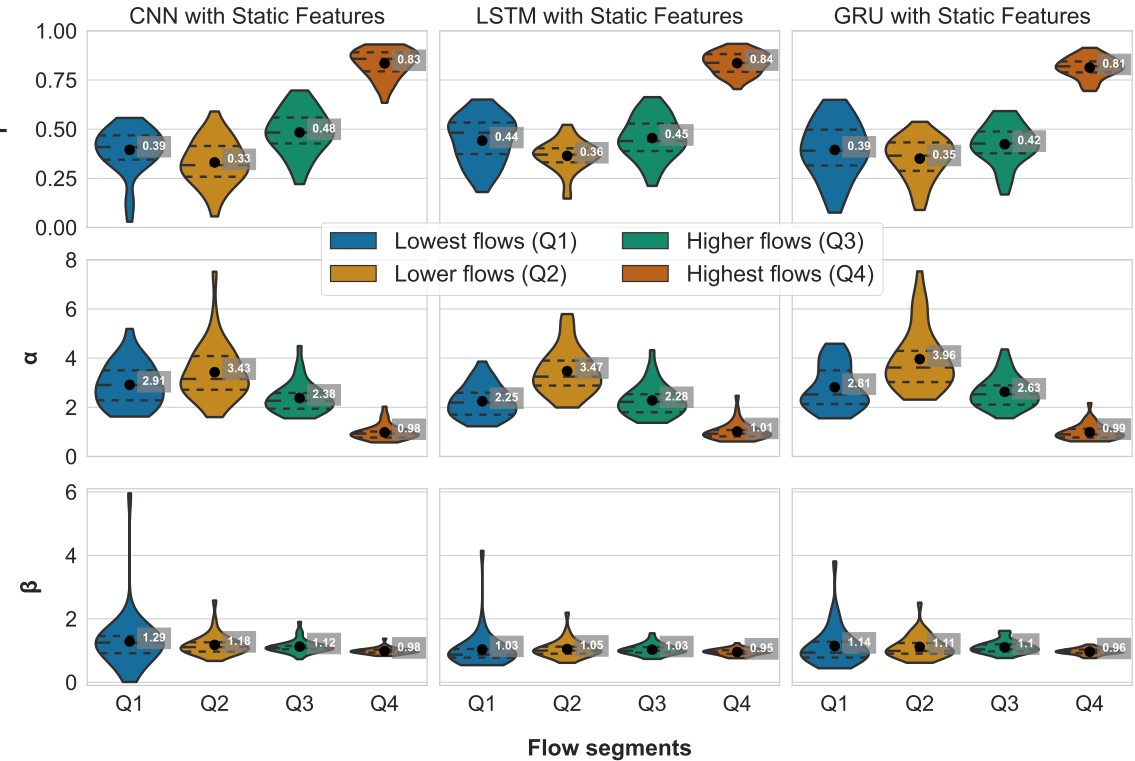

**Figure 8.** Components of the Kling-Gupta Efficiency (KGE) for the employed CNN, LSTM, and GRU models with a batch size of 256 incorporating static features, evaluated across four flow segments: lowest flows (Q1), lower flows (Q2), higher flows (Q3), and highest flows (Q4). From top to bottom, the rows represent the Pearson correlation coefficient (r), the variability ratio ($\alpha$), and the bias ($\beta$). Each violin plot illustrates the distribution of these metrics for all evaluated catchments within each flow segment, with black dots indicating the mean values for each segment. The ideal value for all three metrics is 1, indicating perfect performance.

### 3.3 Model Sensitivity

To elucidate the effect of input parameters on discharge prediction, a thorough sensitivity analysis was conducted. For that, each input feature was uniformly altered and subsequently, the prediction was executed again with the modified inputs. The newly predicted discharge values were then systematically averaged over both time and all catchments. Metric features were subject to an increase of 10% from their original values, while categorical features were configured to dominate across the catchments. Variations in the mean discharge resulting from these adjustments yield insights into the comparative significance of each feature within the model. The results of this analysis are shown in Figure 9, 10 and 11. To enhance visualisation,





the related input features were systematically classified into five distinct groups, namely catchment characteristics, land use, geology, meteorology and soil. Catchment characteristics comprise static features related to the shape, size or elevation of catchments. The land use group remained as it was before. Geology includes the geology type and the permeability of the aquifer. Meteorology encompasses precipitation and evapotranspiration (daily sum and annual mean), while soil combines soil type, soil texture, soil depth and soil temperature (daily mean).

### 3.3.1 CNN

The meteorological features, specifically precipitation and average precipitation, exhibited the most positive impacts on the model, with changes of 11.1% and 8.5% respectively (Figure 9). This underscores their pivotal role in influencing the output of the CNN model. Increasing the daily feature soil temperature led to a decline in the discharge of -2%, likely related to increasing atmospheric water losses with rising temperature through increasing actual soil evaporation and plant transpiration. The static feature average annual mean evapotranspiration had a highly negative influence, causing a reduction in discharge by -8.3%. This demonstrates that not only daily features play an important role in the prediction of the model, but also annual average climatic catchment conditions. The daily forcing evapotranspiration showed a positive impact of 0.4%. The observation that daily evapotranspiration increases with discharge is semingly counterintuitive. However, daily evapotranspiration derived from Jehn et al. (2021) represents actual evapotranspiration, which can increase with wetter conditions and therefor also correlate positively with discharge. In the catchment characteristics category, the elongation ratio and soil depth demonstrated negative effects, causing reductions of -0.3% and -0.7%, respectively. The catchment size and average slope had minor effects, with positive changes of 0.01% and 0.8%, respectively. The minimal influence observed in response to alterations in catchment size is attributable to the standardisation of all water flows, including discharge, to millimeters. Among the categorical features, changing the dominant land use to grassland led to a 2.1% increase in discharge. In contrast, agriculture and forest showed reductions of -4.9% and -1%, respectively. These findings indicate that land use profoundly affects the discharge predictions of the model. However, it is certainly arguable that an increase in agricultural land corresponds to a decrease in discharge.

All soil textures such as 'sandy loam', 'silt loam', 'loam to sandy loam' and 'silty clay' led to reductions in discharge ranging from -0.7% to -3.6%. The observed hierarchy, wherein the discharge decreases progressively from 'sandy loam' to 'silty clay', is coherent and attributable to the reduction in the sand fraction. This decline correspondingly elevates in soil water retention, as shown by Easton and Bock (2021). An exception to this pattern is the class 'loam to sandy loam', which, based on the preceding assumption, should fall within the range between 'sandy loam' and 'silt loam'.

In examining the impact of soil type on discharge, it is observed that all soil categories exhibit a diminishing influence. Specifically, the categories encompassing 'eutric cambisols, stagnic gleysols', 'haplic luvisols, eutric podzoluvisols, stagnic gleysols', and 'eutric cambisols' depict a slight reduction in discharge ranging between -0.7% to -0.9%. Conversely, the 'dystric cambisols' and 'spodic cambisols' classifications demonstrate a more pronounced negative shift, with decreases of -1.7% and -1.8%, respectively. The disparate effects of soil type classes on discharge are complex and do not appear to adhere to a discernible logic. While an increase in 'stagnic gleysols' might logically lead to greater discharge, owing to a potentially





higher rate of surface runoff, the underlying reasons for other relationships are less clear. For instance, it remains ambiguous why an increase in 'spodic cambisols' or 'dystric cambisols' would decrease discharge more whereas 'eutric cambisols' would exert only a small decreasing influence on discharge.

Geological features such as permeability showed positive impacts for rather higher permeable aquifers, ranging from 0.7% to 2.5%. On the other hand, low permeable aquifers such as 'low permeable' and 'low to very low permeable' showed a negative change of -1.4% and -4.4%, respectively. Such findings align with the anticipation that enhanced permeability, and thereby increased hydraulic conductivity, typically result in a more rapid response in groundwater discharge into streams following a precipitation event. However, the hierarchy of the categories does not follow a structured sequence.

Igneous rock types exhibit a negative effect, with a decline of -1.2%, while sedimentary rock types reveal a negative impact of -0.6%. This is congruent with the accepted geological understanding that igneous rock types are often less permeable than sedimentary ones, logically reducing discharge (Jasim et al., 2018). The relatively low impact of the dominant geology type classified as sedimentary can be attributed to the fact that 80% of the catchments fall under this geology type, making the change comparatively minor. Future analyses could address the varying magnitudes of occurrence within the classes by relating the

absolute values of each class to the total occurrences.

    The CNN model demonstrates considerable strength in effectively capturing and leveraging meteorological features, such as precipitation, to improve discharge predictions. However, the model exhibits counterintuitive results with daily evapotranspiration and land use features, indicating potential limitations in handling complex environmental inputs.

### 3.3.2   LSTM

Analogous to the findings from the CNN model analysis, the LSTM model further corroborated that meteorological parameters, namely precipitation and average precipitation, exert the most substantial positive impacts on discharge, registering enhancements of 15% and 11.5%, respectively (Figure 10). Conversely, evapotranspiration and average evapotranspiration negatively impacted discharge, resulting in decreases of -2.2% and -9.7%, respectively. In comparison to the CNN model, the LSTM model displays a substantially higher sensitivity to precipitation, implying that this feature serves as the principal driving force

for this model. Unlike the CNN model, all meteorological features of the LSTM model align with anticipated behaviour, consistent with conventional understanding of hydrological processes. Under catchment characteristics, the elongation ratio showed a minor positive impact on discharge with 0.4%, while the size of the catchment maintained a negligible effect, exhibiting a positive alteration of 0.1%. The average slope resulted in a slight increase in discharge by 0.5%.

    The results of the land use category differed from those of the CNN model. Changing the dominant land use to agriculture

increased the discharge by 1.8%, while more grassland led to a 1.6% increase. In contrast, when forests dominate, there is a decrease in discharge by -1.7%. This outcome not only corresponds to the expected direction of change but also aligns with the hierarchical ordering of the effects associated with land use change.

Contrary to the findings from the CNN model, within the soil type features, the categories 'spodic cambisols', 'haplic luvisols, eutric podzoluvisols, stagnic gleysols', and 'eutric cambisols, stagnic gleysols' exhibited an increasing impact on discharge,





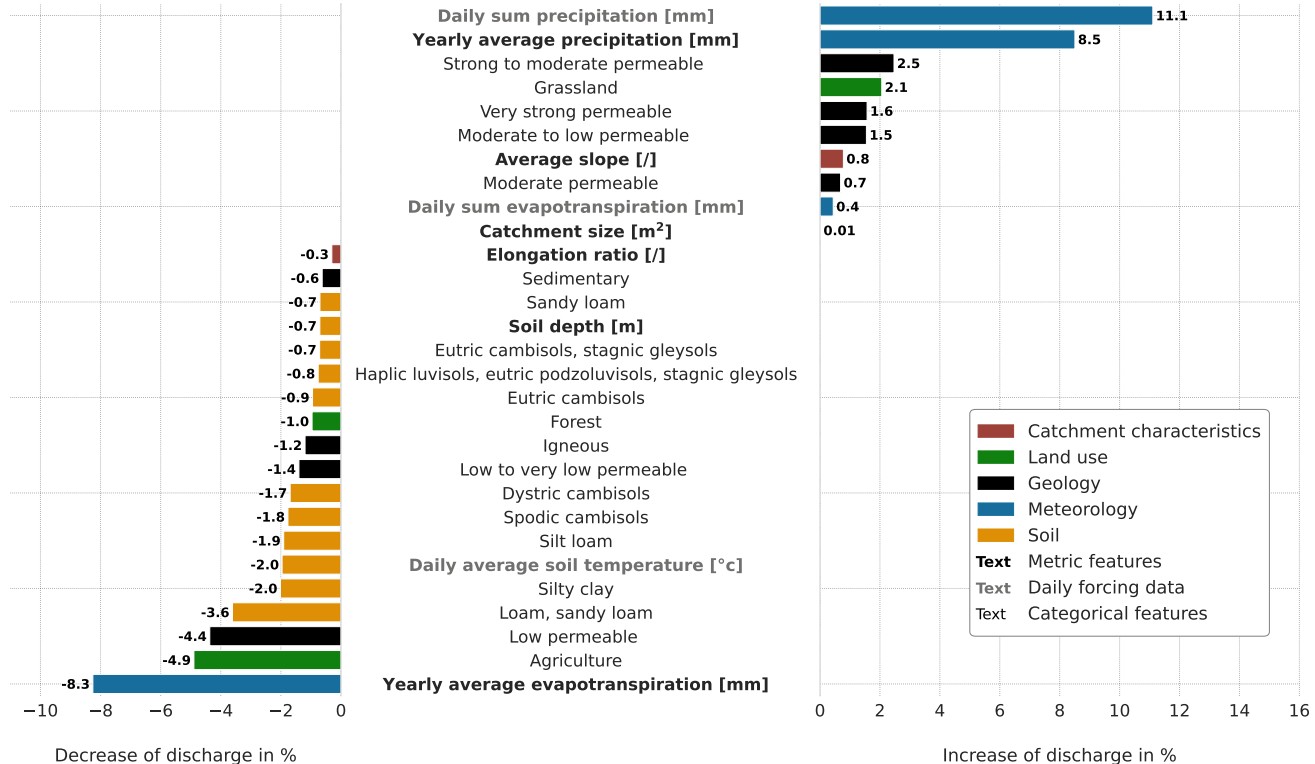

**Figure 9.** Sensitivity analysis of the CNN model with static features and a batch size of 256. All features have been uniformly altered to evaluate their impact on discharge prediction. Metric features were increased by 10%, while categorical features were set as dominating over all catchments.

showing increments of 3.2%, 2.5%, and 1.1% respectively. Moreover, the arrangement of these categories diverges markedly from that observed in the CNN model but lacking an interpretable sequence as well. The other two categories, 'eutric cambisols' and 'dystric cambisols', revealed a decreasing effect on discharge, with decreases of 1.1% and 4%, respectively. It is noteworthy that, while soil types exerted a relatively minor impact on discharge predictions in the CNN model, their influence within the LSTM model is considerably more pronounced, both positively and negatively, with 'dystric cambisols' emerging as the second most significant factor in reducing discharge.

All soil texture categories, with the exception of 'silty clay', which exhibited a decreasing effect of -3.2%, showed an increasing impact on discharge, with variations ranging from 2.5% to 7.2%. This contrasts with the CNN model, where all soil type categories were associated with a decreasing impact. The increase in discharge for 'loam to sandy loam' at 7.2% and for 'silt loam' at 5% signifies a considerably greater influence compared to that observed in the CNN model. The causative factors behind this distinct behaviour remain ambiguous and are likely not solely attributable to soil type characteristics. The daily feature soil temperature revealed a decrease of -3.3%, while soil depth was attributed a marginal impact of -0.2%.





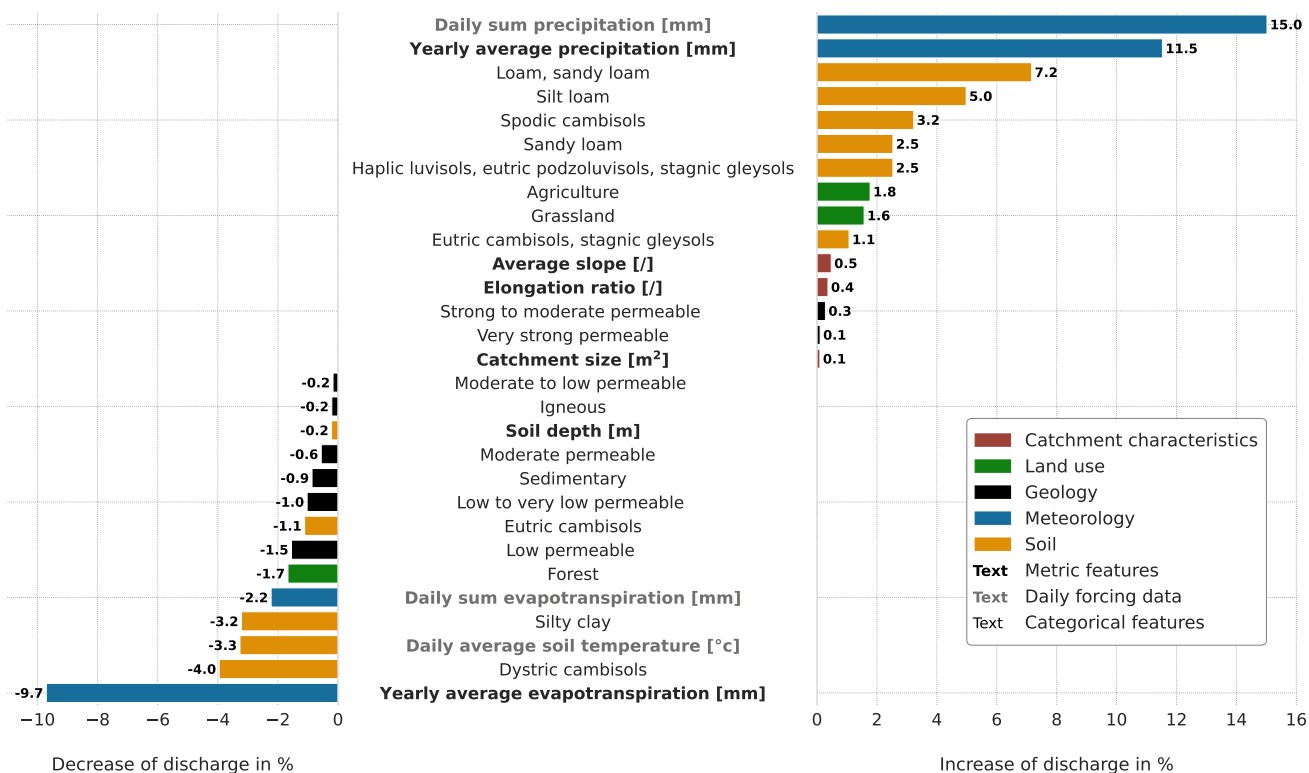

**Figure 10.** Sensitivity analysis of the LSTM model with static features and a batch size of 256. All features have been uniformly altered to evaluate their impact on discharge prediction. Metric features were increased by 10%, while categorical features were set as dominating over all catchments.

Within the geological group, permeability exhibited results similar to those observed in the CNN model, with higher permeable aquifers showing positive effects and moderate to low permeable aquifers demonstrating negative effects. However, the magnitude of increase for the categories with increasing permeability was minimal, ranging only from 0.1% to 0.3%. Only the 'low permeable' and 'low to very low permeable' aquifers displayed a considerable impact, with changes of -1% and 1.5%, respectively. Compared to the CNN model, the role of aquifer permeability in influencing discharge was found to be more pronounced. Igneous rock types showed a marginal negative influence on discharge with -0.2%, whereas sedimentary rocks led to a larger reduction in discharge, amounting to -0.9%. This observed pattern deviates from the anticipated behaviour, given that igneous rocks, typically less permeable than sedimentary rocks, are expected to enhance water retention.

Overall, the sensitivity analysis of the LSTM model revealed a more realistic representation for some features comparted to the CNN model, particularly for land use. However, features like geology type and soil type are better interpreted by the CNN model.





### 3.3.3 GRU

The sensitivity analysis of the GRU model underscores the nuanced response to various hydrological features, drawing parallels
and contrasts with the LSTM model. Notably, meteorological metrics like precipitation and average precipitation exerted strong
positive effects on discharge, with increases of 13.3% and 13%, respectively (Figure 11). In contrast, evapotranspiration and
average evapotranspiration demonstrated negative impacts, decreasing discharge by -3.1% and -9.4%. This heightened sensi-
tivity to precipitation metrics, akin to the LSTM model, further underscores the pivotal role of precipitation in driving the GRU
model's discharge predictions. All meteorological features in the GRU model exhibited expected behaviours, aligning with
established hydrological principles, a consistency observed in the LSTM model as well. This indicates a robust understanding
of meteorological influences by both models.

Regarding catchment attributes, the GRU model proved to be the least affected by these features. The elongation ratio
exhibited a slight negative impact of -0.3% and the catchment size demonstrated a negligible decrease of -0.06%, establishing
this features as not relevant across all three model architectures. The average slope within the GRU model showed also no
relevance (0.02%).

Examining land use impacts, the GRU model paralleled the LSTM model closely. Increasing agricultural land use led to a
1.8% increase in discharge, while grassland prevalence caused a 1.1% rise. However, forest predominance resulted in a modest
reduction in discharge by -0.3%, marking the lowest extent of this category across all models. Nonetheless, this pattern reflects
at least the general anticipated direction in changes of discharge related to land use changes.

The analysis of soil types indicated that the GRU model exhibited a pattern similar to that of the LSTM model. The categories
'spodic cambisols', 'haplic luvisols, eutric podzoluvisols, stagnic gleysols', and 'eutric cambisols, stagnic gleysols' showed a
more noticeable positive effect. The extent of decreasing categories were also more pronounced, with the exception of 'eutric
cambisols', which displayed an identical decrease of -1.1% as observed in the LSTM model.

Every soil texture category, with the sole exception of 'silty clay', which led to a decrease of -2.2%, demonstrated an
increasing effect on discharge, with variations extending from 3% to 11.6%. Similar to the soil type attribute, both recurrent
models maintained a consistent ranking for the soil texture feature, although the influence tended to be more pronounced in the
GRU model. Soil temperature exhibited a uniform reduction in discharge of -3.3%, whereas soil depth demonstrated a decrease
of -1.1%, surpassing the reduction noted in the LSTM model.

In the context of geological features, the GRU model's reaction to permeability followed a pattern akin to that of the LSTM
model, with the exception of 'strong to moderate permeable' and 'very strong permeable' aquifers, which not only exhibited a
more marked impact but also in a coherent order. The influence of igneous rock types continued to be positive, at 5.1%, closely
reflecting the LSTM model's findings, while sedimentary rock types demonstrated a negative effect on discharge, at -1.3%.

In summary, the GRU model's sensitivity analysis reveals a high degree of concordance with the LSTM model in terms
of feature influences on discharge predictions. However, notable differences, particularly in the magnitude of certain features,
highlight the distinct response characteristics of the GRU architecture. Nevertheless, the similarity in effects across many





features suggests that GRU models are adept at accurately discerning hydrological processes, despite their simpler architecture compared to LSTM models.

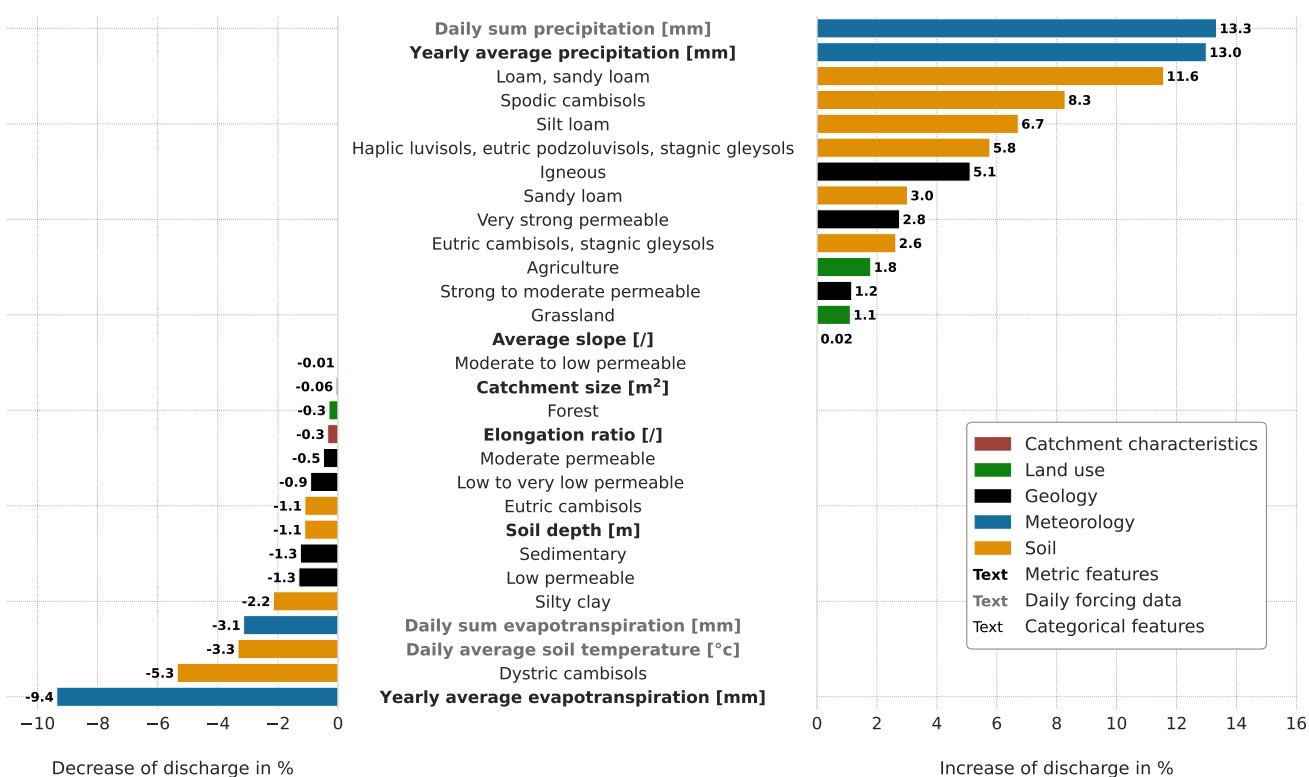

**Figure 11.** Sensitivity analysis of the GRU model with static features and a batch size of 256. All features have been uniformly altered to evaluate their impact on discharge prediction. Metric features were increased by 10%, while categorical features were set as dominating over all catchments.

### 3.3.4 Key Findings

Across all models, meteorological features, particularly precipitation and average precipitation, consistently exhibited sig-
nificant positive impacts on discharge prediction. This underscores the pivotal role of meteorological factors in driving the models' predictions. Conversely, evapotranspiration and average evapotranspiration showed negative impacts on discharge, albeit to varying degrees across the models.

Catchment characteristics, such as elongation ratio and catchment size, had minimal effects on discharge prediction across all models. Land use emerged as an important factor, with changes in dominant land use leading to notable variations in discharge prediction, albeit with some differences in magnitude and directionality between the models.





Soil type and texture displayed varying impacts on discharge prediction across the models, with some categories showing consistent trends while others exhibited divergent behaviours. Similarly, geological features, such as permeability and rock types, influenced discharge prediction differently across the models, highlighting the complexity of their interactions with hydrological processes.

Overall, while there are differences in the sensitivity of the models to certain input parameters, particularly evident in the magnitude of effects, the similarities in the general trends across the models indicate their capability to accurately capture hydrological processes. These findings emphasise the importance of considering various input parameters and their interactions in improving discharge prediction models for hydrological applications.

## 4 Conclusions

This study examined the differences among various neural network architectures, including CNN, LSTM and GRU, in the context of predicting discharge within ungauged basins in Hesse, Germany. The research has shown that all employed ANNs exhibit the capability to accurately discern hydrological processes for discharge prediction over multiple catchments, regardless of the specific architecture. Despite the general use of LSTM models, this study demonstrated that CNN models offer advantages in terms of performance and runtime for time series prediction. In particular, a CNN model showed the highest per-

formance (KGE=0.8), followed by a LSTM model (KGE=0.78) and the GRU model (KGE=0.77). The GRU model generally showed a slightly lower performance with regard to most evaluation metrics. However, given the fact the performance gap is relatively small and that the runtime of the GRU model is 41% faster than the CNN and 59% than the LSTM model, it becomes clear that GRU mode offers a promising balance between predictive accuracy and computational demand. This advantage in runtime becomes particularly salient when dealing with high–resolution time series or when predictions are required on an

extensive scale. Conversely, the examination of the flow segment performance distribution revealed that the LSTM model exhibits superior generalization capabilities across the entire spectrum of flow data, rather than disproportionately depending on peak flow events.

The conducted sensitivity analysis offered valuable insights into the interpretability of the models, revealing that all employed model architectures predominantly provide an authentic representation of the influence of input features. Among these,

meteorological variables, particularly precipitation and evapotranspiration, are pivotal in influencing discharge predictions across all models. This underscores the critical role of accurate meteorological data and the importance of feature selection in enhancing model performance. The impact of land use and soil characteristics on discharge predictions further emphasises the necessity of integrating diverse hydrological and environmental variables to capture the complex dynamics of catchment hydrology.

The results of this study lend additional support to the propositions made by Kratzert et al. (2019a), which advocate that the incorporation of static features can enhance the efficacy of ANNs. Additionally, the relationship between batch size and runtime exhibited distinct variations across the examined models, highlighting the complex interplay between architectural





design and hyperparameter configuration. However, an increase in batch size was found to diminish the performance in terms of discharge prediction. Additional exploration may more accurately assess the impact of varying batch sizes by maintaining a

consistent set of hyperparameters while altering the batch size.

These insights not only serve as guidance for researchers utilising neural networks in hydrology but also contribute to a comprehensive framework for evaluating different algorithms. Furthermore, this research bridges a critical gap in hydrological modelling literature by systematically comparing the efficacy of different neural network architectures in predicting discharge in ungauged basins, thereby paving the way for more informed and effective application of artificial intelligence in hydrology.

Future research may delve into the exploration of other neural network architectures and techniques, such as transformer models. In summary, successful prediction in ungauged basins accentuates the potential of neural networks in the field of hydrology.

*Code and data availability.* The entire code, along with the data sets upon which this study relies, except for the discharge data, can be accessed publicly in the following repository: Neural-networks-in-catchment-hydrology.git.



# Appendix A

## A1 Hydrographs of the CNN model with static features and batch size of 256

### A1.1 Highest performance

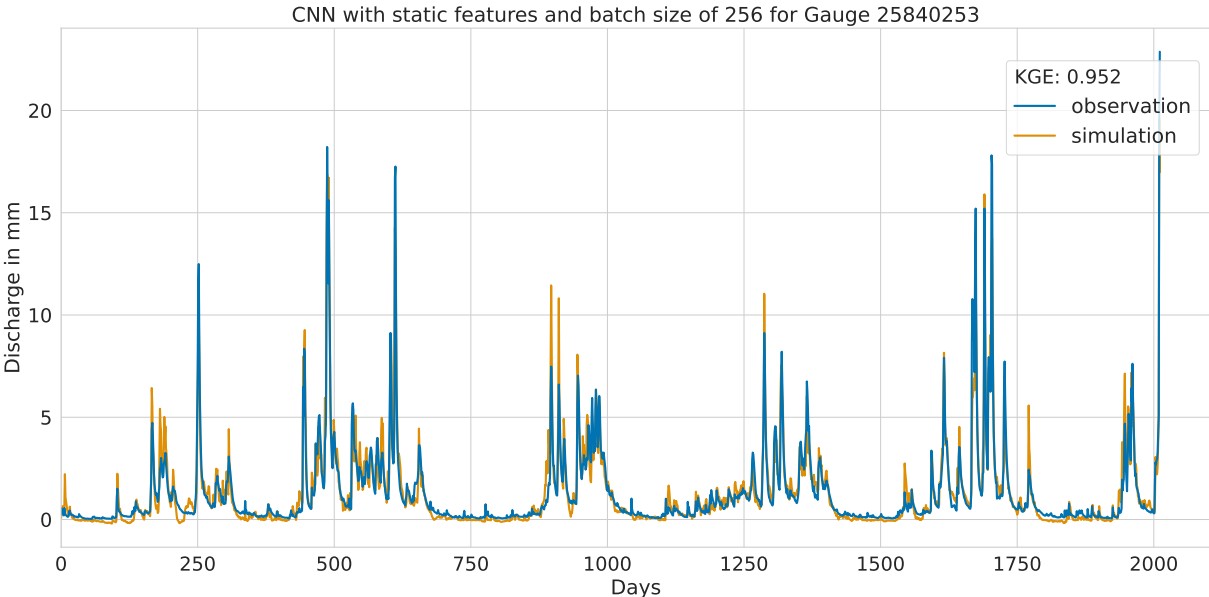

**Figure A1.** Hydrograph of observed (blue) and predicted discharge (orange) with the Kling-Gupta Efficiency (KGE)





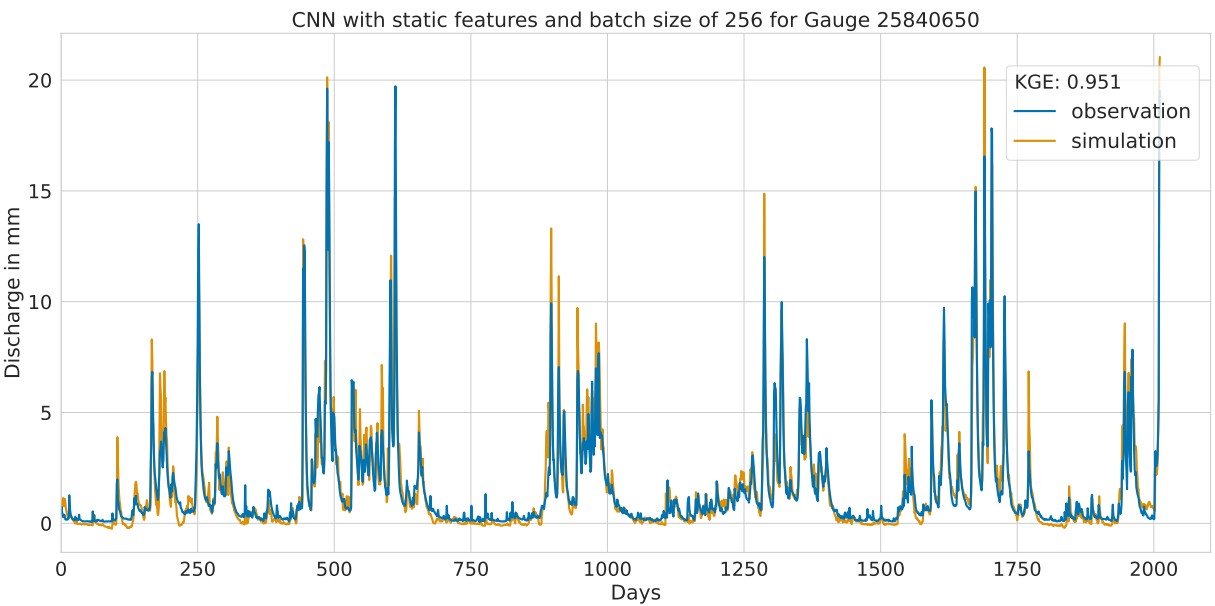

**Figure A2.** Hydrograph of observed (blue) and predicted discharge (orange) with the Kling-Gupta Efficiency (KGE)

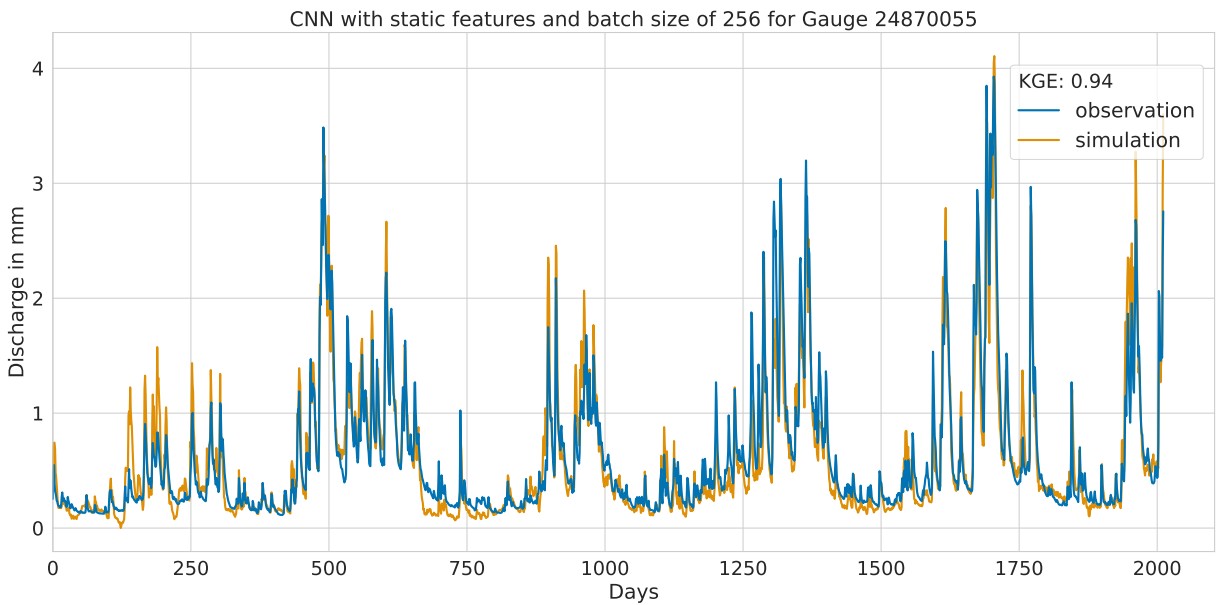

**Figure A3.** Hydrograph of observed (blue) and predicted discharge (orange) with the Kling-Gupta Efficiency (KGE)



## A1.2 Lowest performance

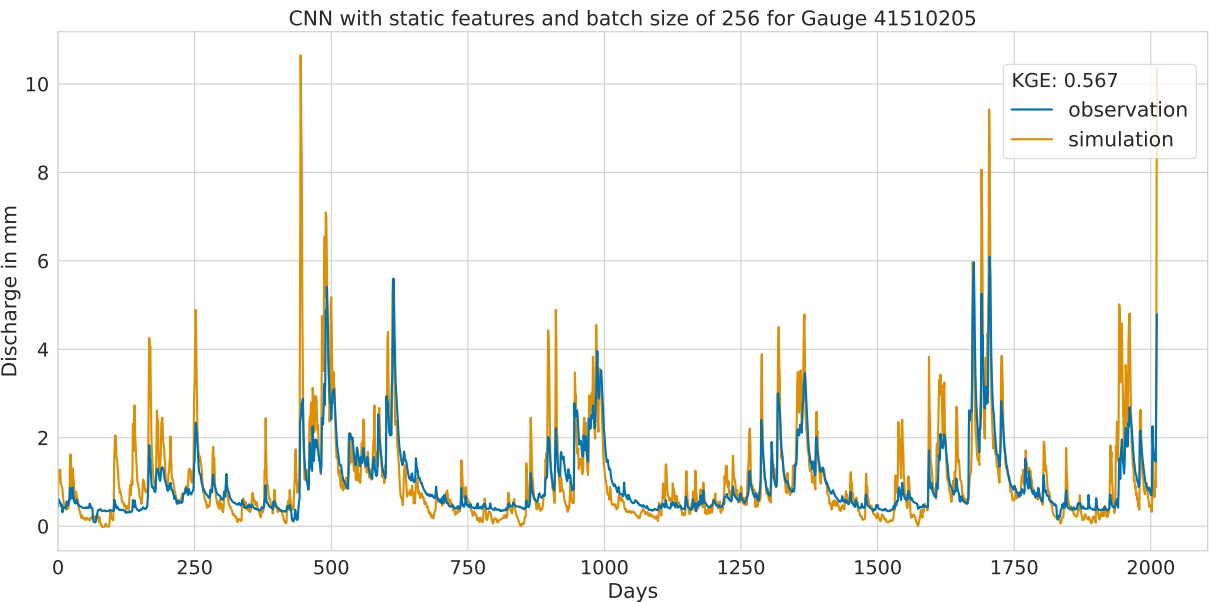

**Figure A4.** Hydrograph of observed (blue) and predicted discharge (orange) with the Kling-Gupta Efficiency (KGE)





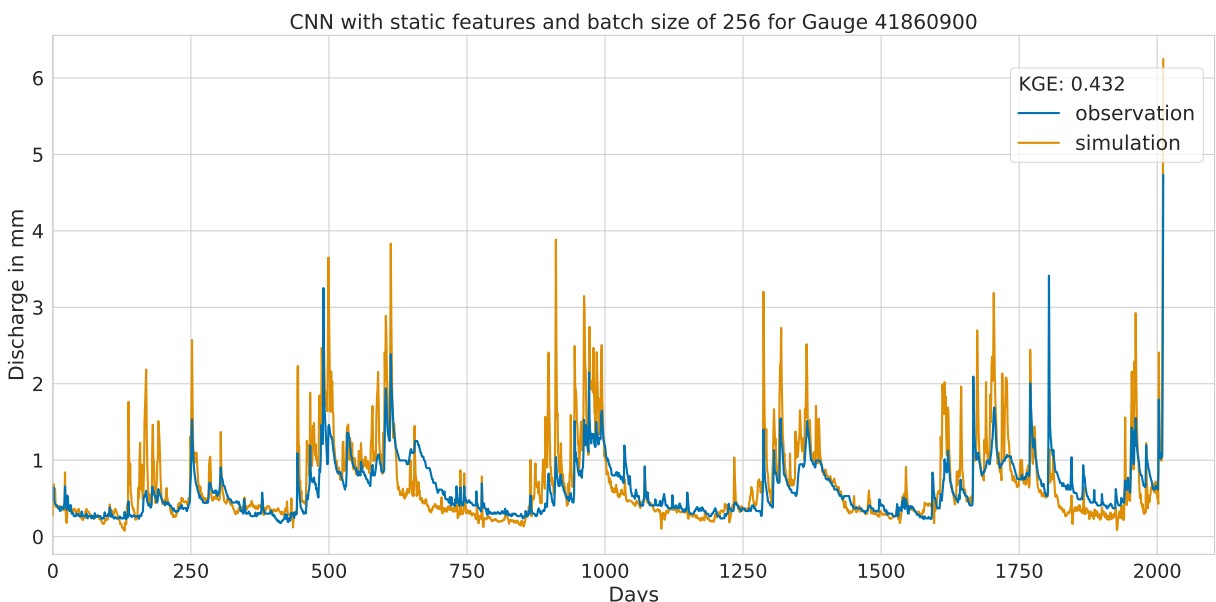

**Figure A5.** Hydrograph of observed (blue) and predicted discharge (orange) with the Kling-Gupta Efficiency (KGE)

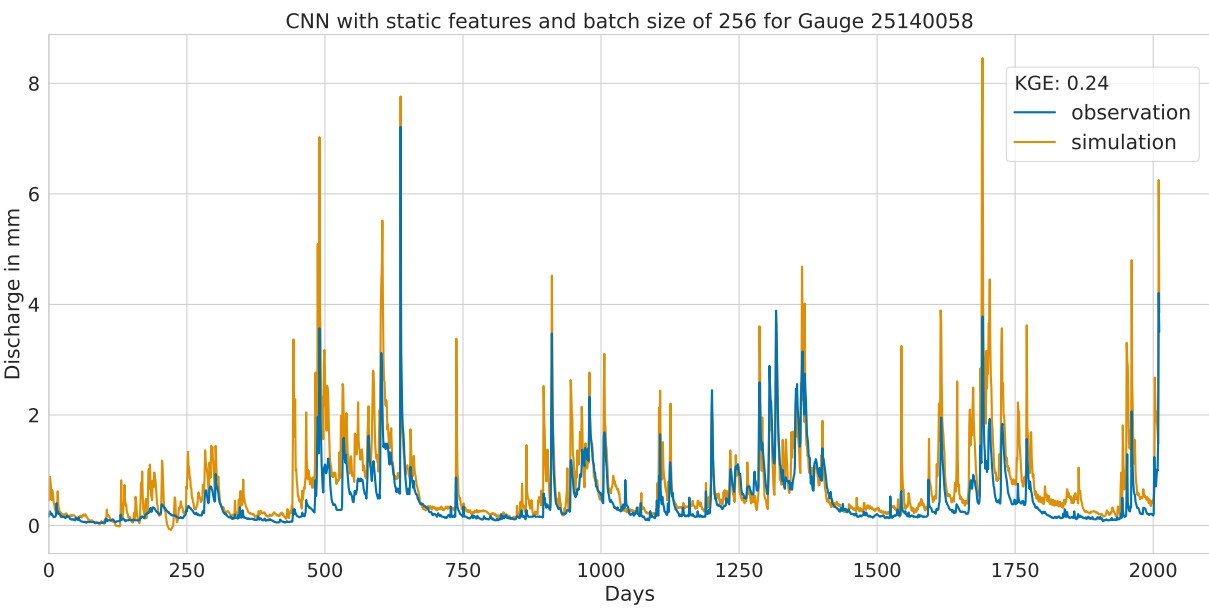

**Figure A6.** Hydrograph of observed (blue) and predicted discharge (orange) with the Kling-Gupta Efficiency (KGE)





## A2 Hydrographs of the LSTM model with static features and batch size of 256

### A2.1 Highest performance

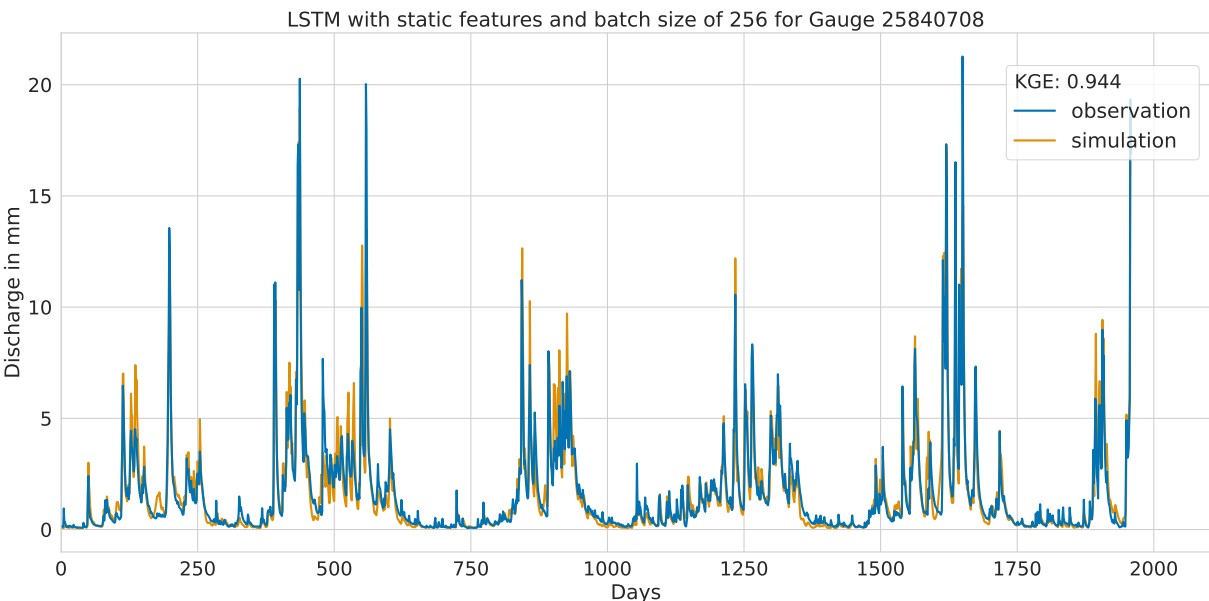

**Figure A7.** Hydrograph of observed (blue) and predicted discharge (orange) with the Kling-Gupta Efficiency (KGE)





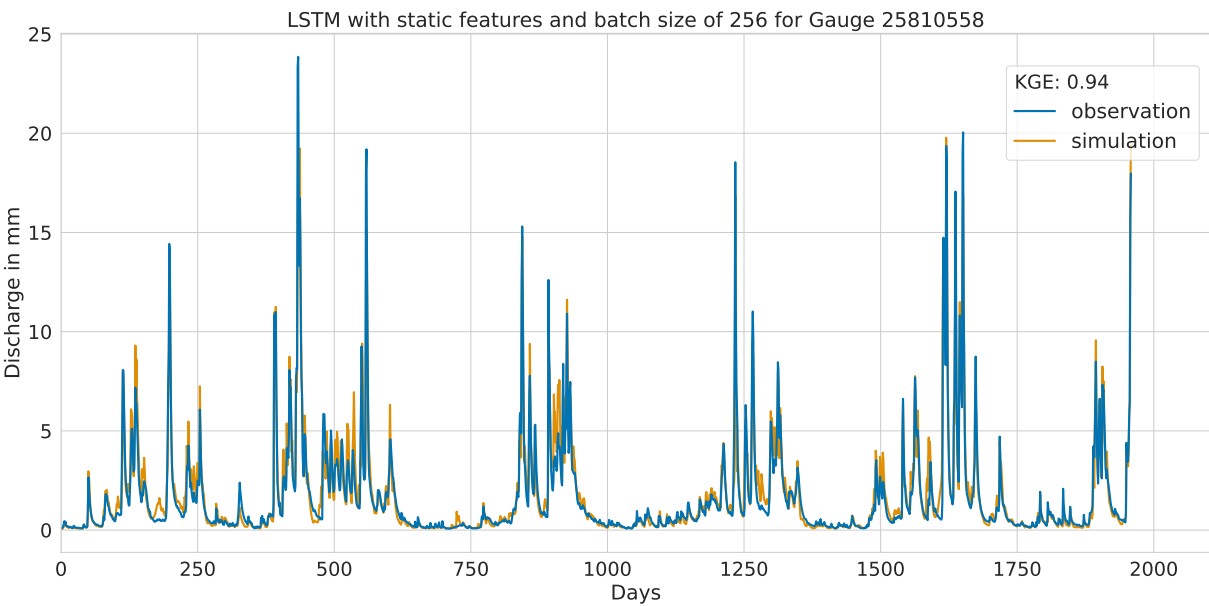

**Figure A8.** Hydrograph of observed (blue) and predicted discharge (orange) with the Kling-Gupta Efficiency (KGE)

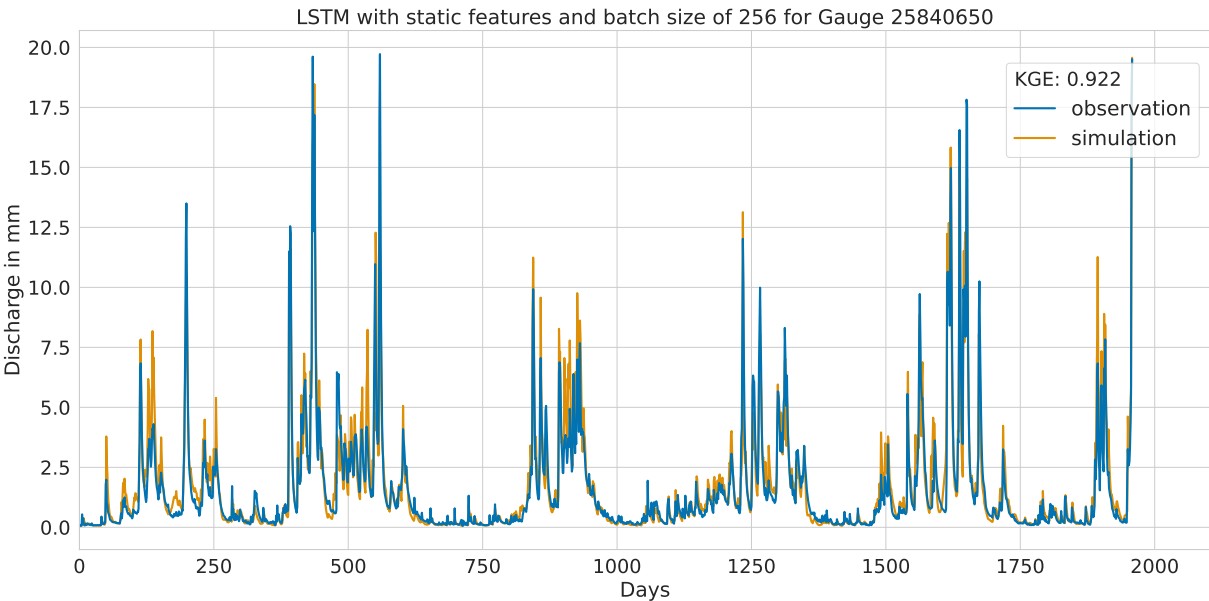

**Figure A9.** Hydrograph of observed (blue) and predicted discharge (orange) with the Kling-Gupta Efficiency (KGE)





## A2.2 Lowest performance

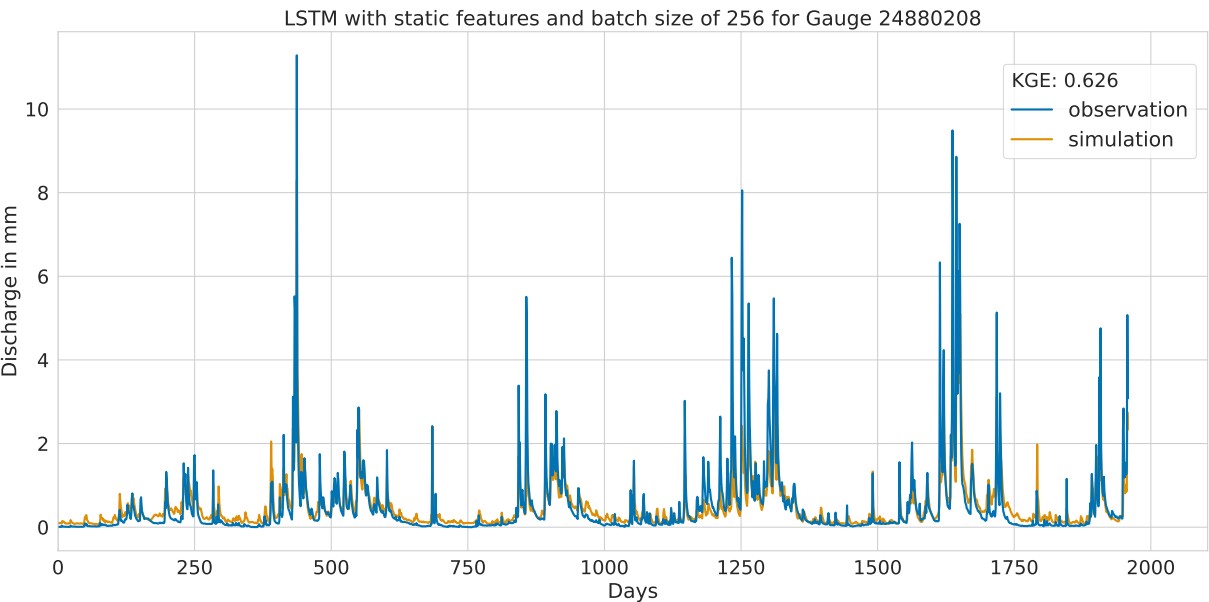

**Figure A10.** Hydrograph of observed (blue) and predicted discharge (orange) with the Kling-Gupta Efficiency (KGE)





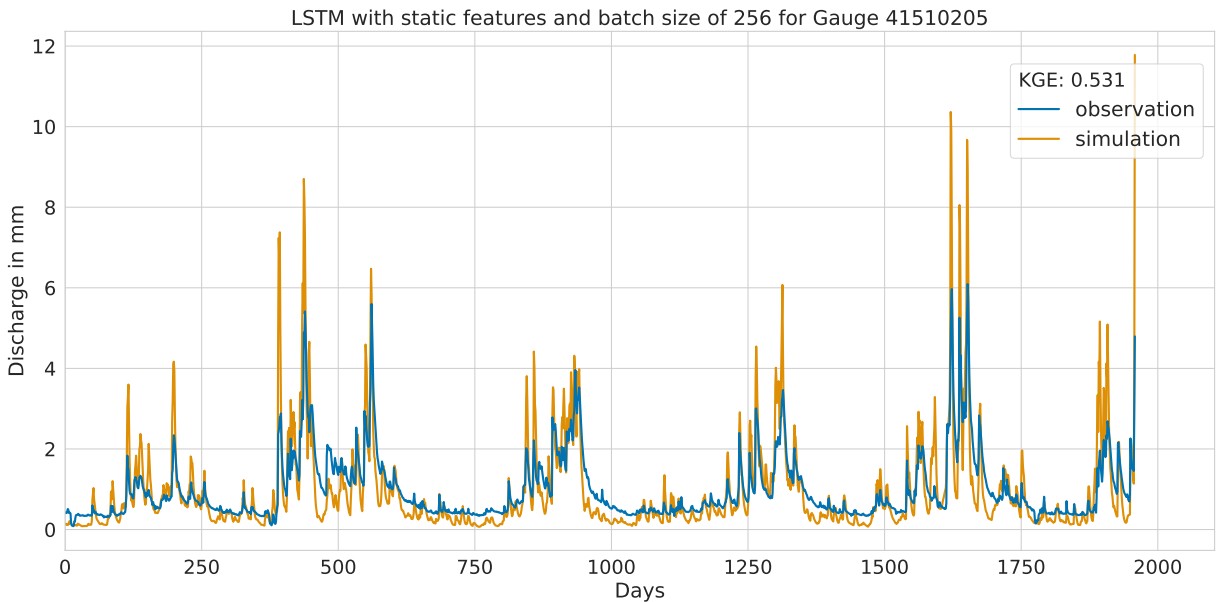

**Figure A11.** Hydrograph of observed (blue) and predicted discharge (orange) with the Kling-Gupta Efficiency (KGE)

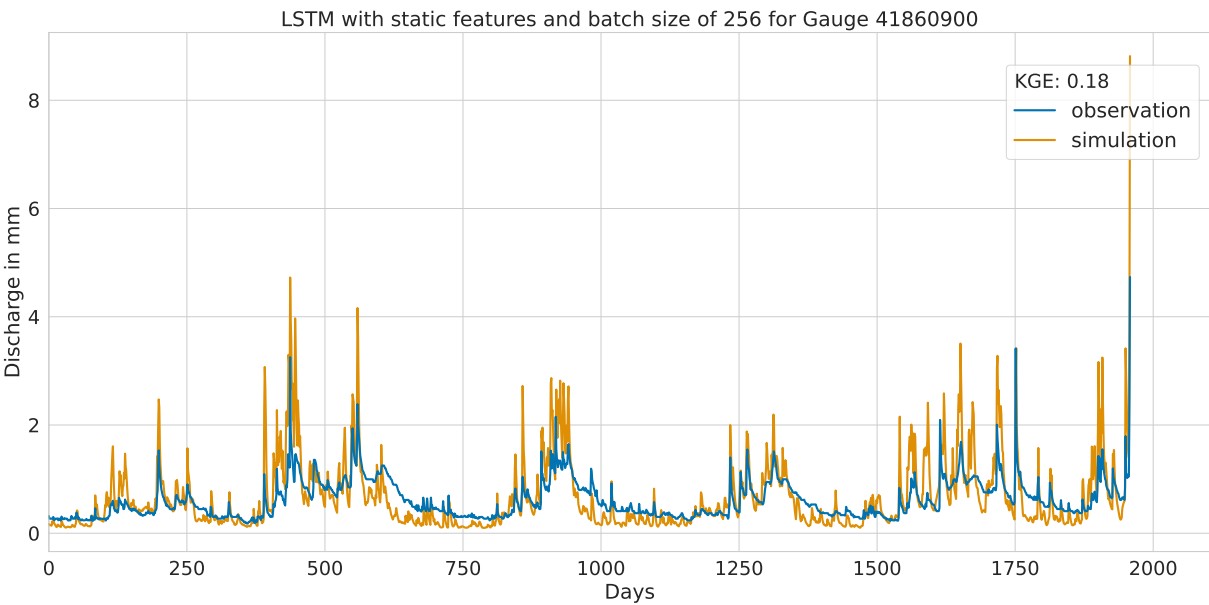

**Figure A12.** Hydrograph of observed (blue) and predicted discharge (orange) with the Kling-Gupta Efficiency (KGE)





## A3  Hydrographs of the GRU model with static features and batch size of 256

### A3.1  Highest performance

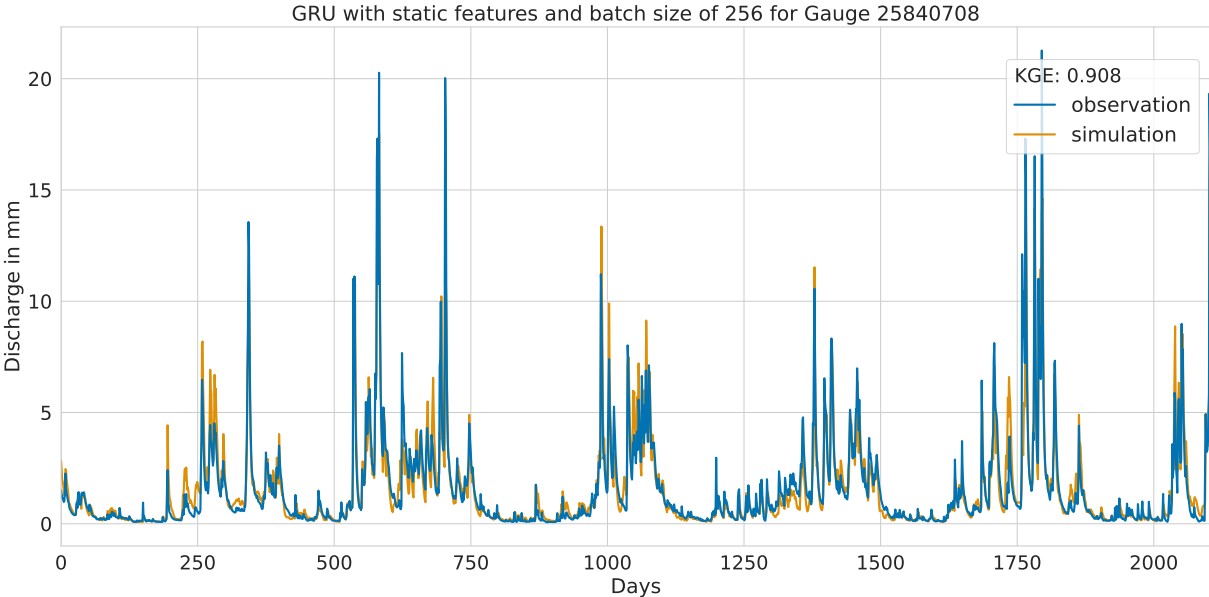

**Figure A13.** Hydrograph of observed (blue) and predicted discharge (orange) with the Kling-Gupta Efficiency (KGE)





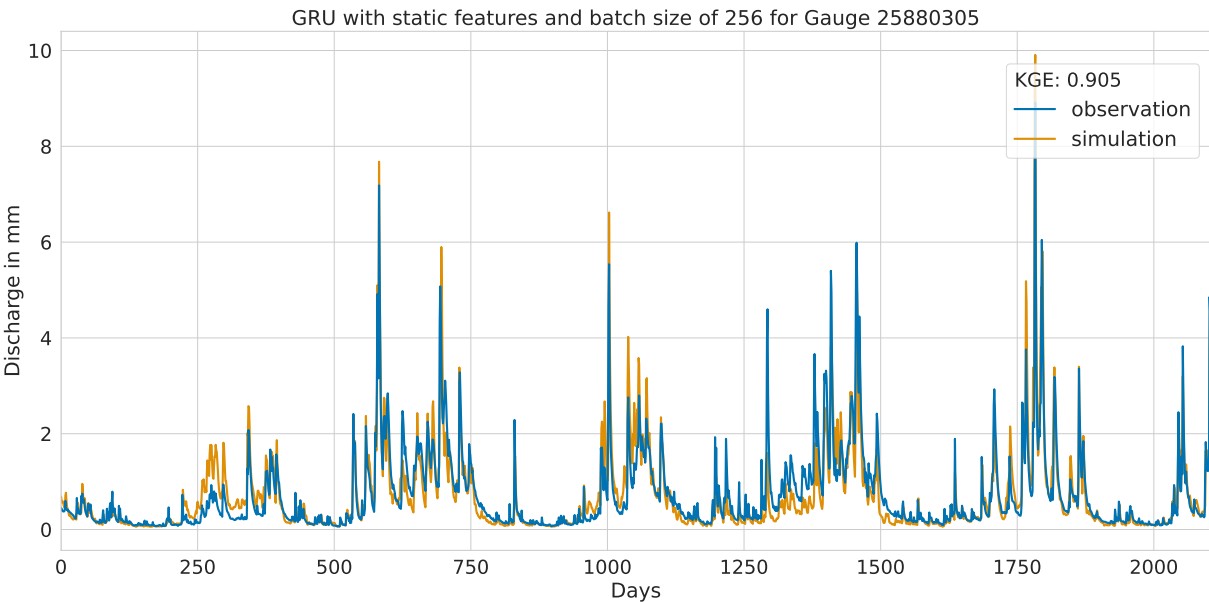

**Figure A14.** Hydrograph of observed (blue) and predicted discharge (orange) with the Kling-Gupta Efficiency (KGE)

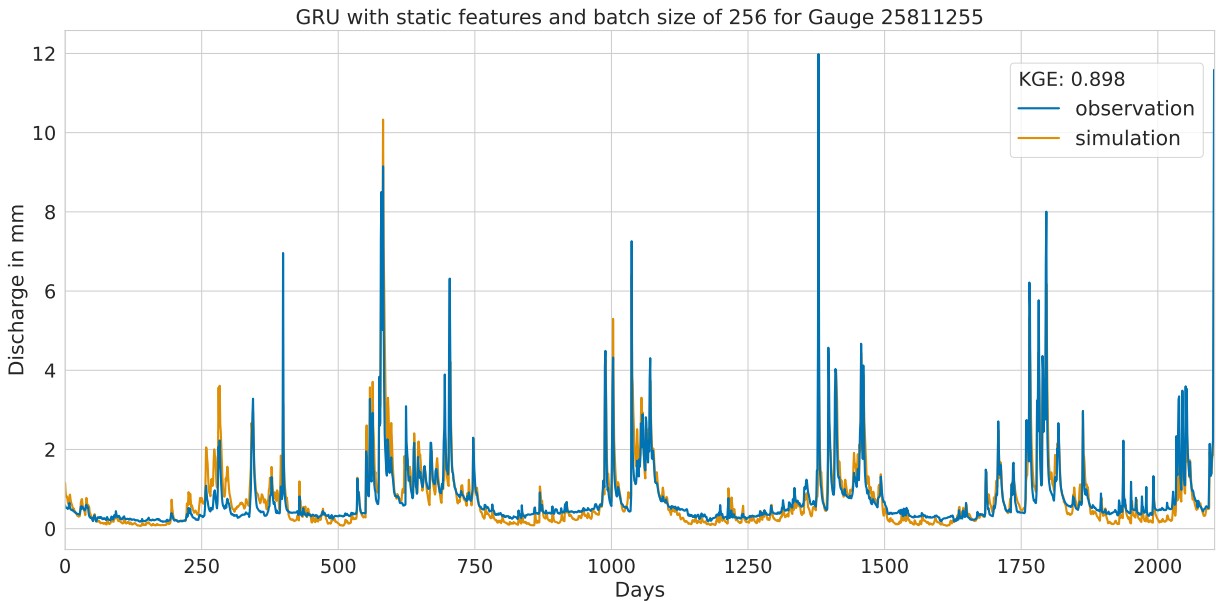

**Figure A15.** Hydrograph of observed (blue) and predicted discharge (orange) with the Kling-Gupta Efficiency (KGE)





## A3.2 Lowest performance

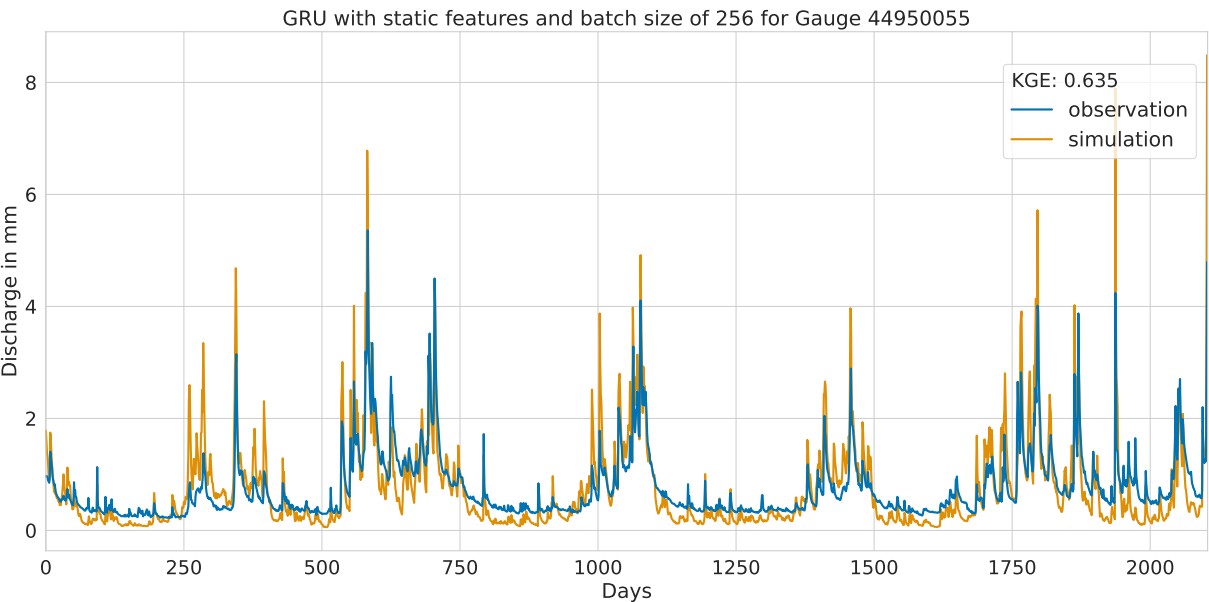

**Figure A16.** Hydrograph of observed (blue) and predicted discharge (orange) with the Kling-Gupta Efficiency (KGE)





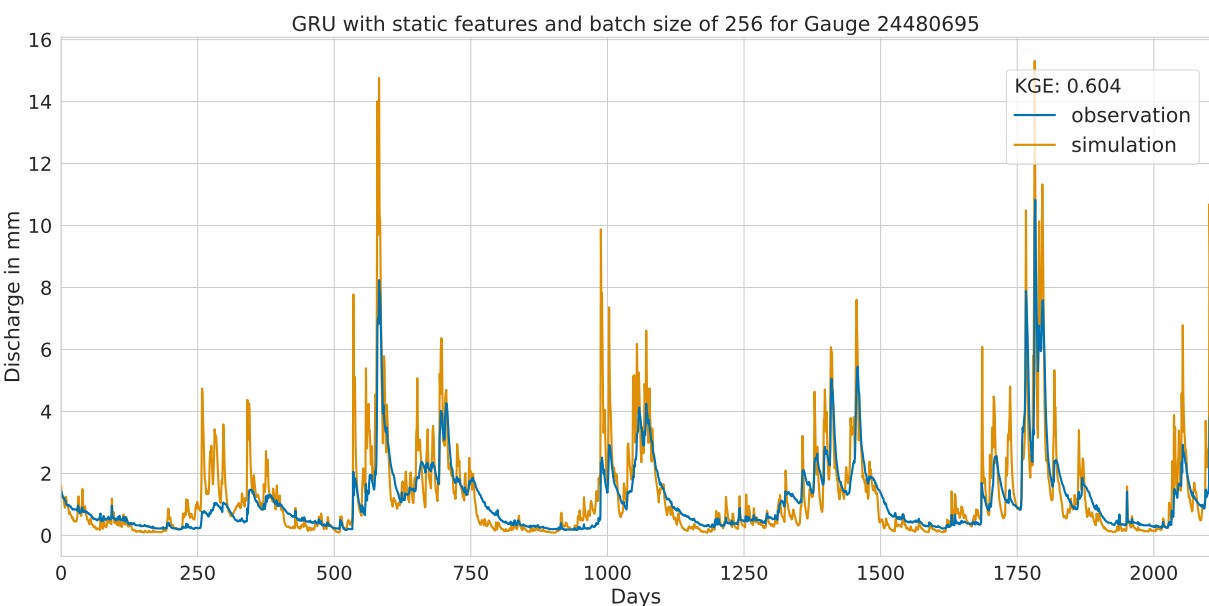

**Figure A17.** Hydrograph of observed (blue) and predicted discharge (orange) with the Kling-Gupta Efficiency (KGE)

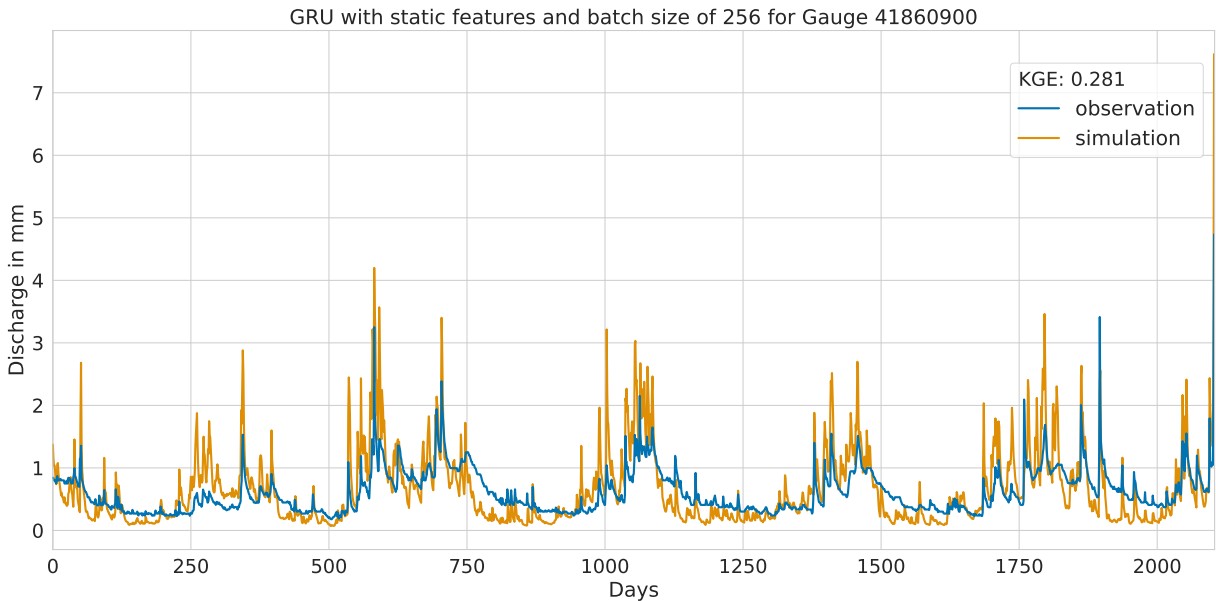

**Figure A18.** Hydrograph of observed (blue) and predicted discharge (orange) with the Kling-Gupta Efficiency (KGE)



*Author contributions.* MW carried out the analysis and wrote the paper. MW developed the model code and performed the simulations. TH and LB reviewed and edited the paper.

*Competing interests.* The contact author has declared that none of the authors has any competing interests.

*Acknowledgements.* I acknowledge the Institute for Landscape Ecology and Resources Management (ILR) for its ongoing support and guidance throughout all the necessary steps.



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
