# Peer review of "Neural networks in catchment hydrology: A comparative study of different algorithms in an ensemble of ungauged basins in Germany"

_Hydrology and Earth System Sciences, 2024_

## Author Response (AR1)

**Revision 1**

**Major comments:**

**Metrics:**

The authors make their analysis by reporting the mean metrics. When reporting results for regional models usually the median is a more robust estimate, because it is influenced less by outliers. That is why studies like Kratzert et al (2019b) https://doi.org/10.5194/hess-23-5089-2019, Lees et al (2021) https://doi.org/10.5194/hess-25-5517-2021 and Feng et al (2022) https://doi.org/10.1029/2022WR032404 (among others) report both mean and median or only median. This property of being robust against outliers is especially important when one is comparing the different models because if one is reporting only the mean, one really bad-performing catchment can affect the overall mean value and the relative ranking between the models might change. Is there a reason why the authors prefer to use the mean? I am not saying that the authors need to change their reported metric, maybe they can just calculate the median for a couple of metrics to see if the relative ranking between the models stays the same. I leave it to the discretion of the authors and the editor to see if this change is necessary.

This is a really good point, and there is no specific reason for choosing only the mean values within our study. We added the median to table 5 as a more robust metric in the revised version of the manuscript to support a better comparison of the models.

**Statistical-significant:**

The differences between the overall model performance when considering static features (Figure 5) are quite small (CNN: 0.8 – LSTM:0.78 – GRU:0.77). You are reporting in line 317 that the CNN is slightly better than the others. Are these differences statistically significant? A quick test that can be made is to train a small ensemble (3-5 models) with different random initializations, to see if the differences are just due to the stochastical nature of the training process or are indeed statistically significant. I leave it to the discretion of the authors and the editor to see if this change is necessary.

The reviewer has a good point here, a test on significance would be a very good addition, not only to this manuscript, but too a lot of recent publications in this field. However, as it is not our intention to claim at this point that any of tested models is better or worse, we decided to leave this test for future work. However, in the spirit of the reviewers raised point, we decided to call for statistical tests in the discussion section of the manuscript.

The text reads: "Additionally, to better compare the performance differences among the models, multiple runs of each model with different random initializations should be considered, as small observed differences might result from the stochastic nature of model initialization and optimization processes." (L709-711)

**Sensitivity analysis:**

I do have a major concern during the sensitivity analysis. There is no reason why data-driven models should interpret static attributes as we do. When training/testing a regional model, the static attributes help the model contextualize the type of basin it is dealing with. However, I do not think that there is enough information during training to be able to do a proper sensitivity analysis with the static attributes. In this case, the authors have 11 static attributes. This means they have an 11-dimensional space filled with only 54 points. I would argue that it is difficult to infer/capture relevant patterns in these conditions, and this could be the reason why there are confusing patterns in the reported results. For example, in some models, a positive perturbation in a static attribute increases the discharge, while in other models the same effect decreases it. The authors mention this in several cases:

- Line 525: "The disparate effects of soil type classes on discharge are complex and do not appear to adhere to a discernible logic"
- Line 534: "However, the hierarchy of the categories does not follow a structured sequence."
- Line 569: "The causative factors behind this distinct behaviour remain ambiguous and are likely not solely attributable to soil type characteristics"
- Line 524: "Land use emerged as an important factor, with changes in dominant land use leading to notable variations in discharge prediction, albeit with some differences in magnitude and directionality between the models."
- Line 626: "Soil type and texture displayed varying impacts on discharge prediction across the models, with some categories showing consistent trends while others exhibited divergent behaviours"

Therefore, I am not sure that the patterns that are being reported as logical/significant, have an actual causal connection or are associated with other factors. I would suggest that the authors reduce the sensitivity analysis to only perturbations in the dynamic inputs, which I think are quite consistent for most of the cases, and are better justified. However, if there is a reason why the authors consider that the sensitivity analysis of the static attributes should be kept, I will gladly reconsider/discuss it.

Your concern is right. There is not enough data to fully understand the behavior of the models with regard to static features. We might got lost in finding patterns where there is no statistically robust evidence, and therefore decided to remove all parts of the sensitivity analysis with regard to static features.

**Minor comments:**

Line 110: The authors are saying that "encoding accommodates ordinal scales, which is better suited for hierarchical features such as permeability". However, it should also be considered that encoding gives a sequential order to classes in which a

sequential order does not make much sense (soil type, soil texture...). It is ok to use encoding, this does not need to be changed, but you should also indicate the disadvantages.

We agree with the reviewer. The text reads now "Moreover, label encoding accommodates ordinal scales, which is better suited for hierarchical features such as permeability. In contrast, categorical features without a meaningful order, such as soil type or soil texture, are better handled by one—hot—encoding, which treats each category independently" (L109-113)

Line 114: Is there a reason the min-max transformation was used instead of standardization?

In our previous trials, the min-max scaler showed better results, so we kept using it. As the focus of this manuscript is not on the transformation approach, we decided to inform the reader about the approach used and do not show any comparison.

Line 121: At first it is not clear the purpose of the moving window. If I understood correctly, it is to create the batches. If so, please indicate this at the beginning.

We agree with the reviewer. The text reads now "To transform the data sets into training batches a two-dimensional moving window, characterized by dimensions T × D, was subsequently implemented, where T represents the moving window size, also known as look-back period or sequence length (Figure 2)." (L123-125)

Line 142: What do you mean by "with the exception of the recurrent layer"?

We meant that the actual LSTM and GRU layer within our models, which inherits the recurrent calculation, is the only difference between the two models. The text reads now "Because the employed LSTM and GRU models possess an identical layer structure, both models share an equivalent set of hyperparameters." (L145-147)

Line 221: "The loss function is regulated by an algorithm known as the optimizer." Can you further elaborate here? Because I would argue that the opposite case is true. The optimizer is regulated by the loss function.

The reviewer is right and this error has been corrected in the revised version of the manuscript. The text reads now "The optimizer, an algorithm designed to minimize the loss, regulates the process of updating the model's parameters. This optimizer strives to enhance the model performance by iteratively determining the loss and then adjusting the model parameter to reduce this loss" (L225-226)

Line 224: "Thus, by minimising the loss, the machine learning model can improve its predictive accuracy and thereby enhance its capacity to generalise from the training data to unseen instances". I do not agree with the second part of this sentence. By minimizing the loss, you indeed improve the predictive accuracy, but this action by itself does not assure you good generalization capabilities. If you overfit your model,

you will reduce the training loss as much as possible, but you will lose the generalization capabilities of the model.

The reviewer is right, we changed the sentence to "Thus, by minimizing the loss, the machine learning model can improve its predictive accuracy" (L229-230).

Line 226. "The optimizer used for all utilised models in this study is the Adam–optimizer." It would read better if you removed utilized. Also, would be better to cite the paper here instead of the next line.

**Changed as proposed.**

Line 237: In line 115 you indicate that the target is also scaled. If you scale your target data, then the direct evaluation of certain facets of the discharge time series by the KGE is no longer applicable (see section 6.3 of Santos et at., 2018). If this is the case, please modify this sentence.

The study of Santos et al. 2018 states that when using logarithmic transformation it can come to issues. But since we use a linear scaling method (min-max scaler) and reversed the scaling before calculating the KGE, there should be no issues.

Line 272: What is the purpose of increasing the learning rate in the first 3 epochs?

The purpose of the warmup period is to allow the model to explore the parameter space with a smaller learning rate initially, before transitioning to the main learning rate schedule. This strategy helps to prevent large fluctuations in the loss function during the early stages of training and facilitates a smoother optimization process.

https://doi.org/10.48550/arXiv.1812.01187 https://doi.org/10.48550/arXiv.1706.02677

Line 278: "The analysis depicted in Figure 5 delineates a comparative evaluation of model efficiency..." Figure 5 indicates model performance, not model efficiency.

**Changed as proposed.**

Line 339: It does not make much sense to compare to Nguyen et al. (2023a) if the conditions are so different. If I understood correctly, they trained single basins LSTM and are not evaluating PUB. This is different from what you are doing. The other comparison by Kratzert makes more sense.

A model that is calibrated to only a single basin tends to give better results than a model that is generalized over several catchments, especially for PUBs. We wanted to show that the model trained in this study shows even better results than the specialized model. We clarified our argumentation: "In the context of existing literature, Nguyen et al. (2023a) reported an NSE of 0.66 for an LSTM model calibrated across three distinct catchments, each with its own separate calibration and not extending to ungauged scenarios. While models calibrated to individual basins often perform better than those generalised across multiple catchments,

particularly in PUB, our results demonstrate that the generalised models trained here achieves even better results than these specialized model" (L351-355)

Line 346: Would be good to write the PBIAS equation. Because depending on how you write it, a positive/negative value can be associated with under/over estimation.

Changed as proposed. For clarity, we added the formula of all used metrics. (L345-348)

Line 507: You state, "The daily forcing evapotranspiration showed a positive impact of 0.4%. The observation that daily evapotranspiration increases with discharge is seemingly counterintuitive. However, daily evapotranspiration derived from Jehn et al. (2021) represents actual evapotranspiration, which can increase with wetter conditions and therefor also correlate positively with discharge." This is the argument that you use to justify the result. On the other hand, when analyzing the other two models, you indicate that the LSTM and GRU produce lower discharge with higher evapotranspiration. In line 550, you mention "all meteorological features of the LSTM model align with anticipated behavior, consistent with conventional understanding of hydrological processes". Similarly, in line 589, you state that for the GRU model "All meteorological features in the GRU model exhibited expected behaviors, aligning with established hydrological principles, a consistency observed in the LSTM model as well." Given that you are using the same input data for all models, the argument for the CNN case becomes invalid. The ET-Q relationship cannot be consistent in the CNN if it is positive and also consistent in the LSTM and GRU if it is negative.

We agree that if the input data is the same, the response of the models should also be the same. However, it appears that the different modelling approaches interpret the input data differently. From our point of view, there is no objective, unambiguous reason to judge one or the other interpretation of the models as 'correct'. Our argumentation therefore explains the possible different behaviours of the models without evaluating them. We have made this clearer in the revised version of the manuscript "Although this may offer a plausible explanation for the observed anomalous behavior, it is unlikely within the context of this study. Given that all models share the same input features, both the LSTM and GRU models should exhibit similar behavior, which is not observed (see Figure 9)." (536-539).

Line 508: Typo: therefore

**Changed as proposed**

Line 517: I do not agree that it is coherent that a higher percentage of sand and a lower percentage of clay should produce higher discharges. Sand has a higher infiltration capacity, and clay has a lower permeability. Therefore, depending on the case, you will have higher discharges with higher content of clay, because less water will infiltrate, and more direct runoff will be produced.

All soil types such as 'sandy loam', 'silty loam', 'loam to sandy loam' and 'silty clay' led to a reduction in runoff of between -0.7 % and -3.6 %. The decrease in runoff from 'sandy loam' to 'silty clay' could be explained by a reduction in the sand content and thus increasing water holding capacity of the soil (Easton and Bock, 2021). However, the lower infiltration capacity of clayey soils compared to sandy soils could also lead to a lower infiltration rate and increased surface runoff. OVerall, the catchments we investigated in Hesse are typically humid mountainous catchments in which surface runoff plays a minor role in runoff generation processes (Jehn et al. 2021, Breuer et al. 2009) and subsurface stormflow dominates (Chifflard et al. 2019). Due to the shortening of the sensitivity analysis, this point is now redundant.

Line 648: "revealing that all employed model architectures predominantly provide an authentic representation of the influence of input features". I am not sure if your data supports this for the static attributes. The different models present different behaviors toward changes in the static attributes.

We changed the statement to cover daily feature only. The text reads now: "The sensitivity analysis provided valuable insights into the interpretability of the models, demonstrating that all model architectures accurately capture the impact of the non-static input features, with the exception of daily evapotranspiration in the CNN model. Precipitation emerged as the most significant driver of discharge predictions across all models. " (L689-697)

**Revision 2**

This paper compared three neural networks: CNN, LSTM and GRU, in discharge prediction in 54 catchments in Hesse, Germany. Detailed model comparison were conducted according to batch sizes, in/ex-cluding static attributes, computational efficiency, as well as model sensitivity. Generally, the structure of the paper is clear and well-organized, however, there are some concerns.

**Specific comments:**

1) For each catchment, it is unclear that authors used only one time series or the spatial averaged time series discharge data and three meteorological factors, as well as for the 11 static catchment features.

We are not sure, what you mean. But we added additional information about the discharge data. The text reads now "For each catchment, daily sum of precipitation [mm], daily sum of evapotranspiration [mm] and soil temperature in 5 cm soil depths [°C] are available along with the corresponding discharge [mm]. The discharge data is obtained from a gauging station located within the respective catchment." (L89-91)

2) In the model training section, it seems that the parameters are set to be same in the three models. However, the CNNs generally need more epochs to converge than recurrent neural networks. Could authors list the specific/optimized parameter values for each model?

**The specific hyperparameter of each model are presented in Table 6.**

3) Could authors elaborate why static catchment attributes improve the overall model performance in line 284?

The text reads now "This aligns with the findings presented by Kratzert et al. (2019b), who assert that static catchment attributes enhance overall model performance by improving the distinction between different catchment-specific rainfall-runoff behaviors" (L288-290)

4) We can not get a conclusion that smaller batch sizes contributed to better predictive performance based on the only comparison between 256 and 2048. More batch sizes should be tested.

In our paper we compared only two batch sizes (small vs large), of which the smaller one showed better performance, hence the assumption. We reformulated the results to "the smaller batch size of 256 contributes to better model performance with regard to mean KGE values." (L322)

5) For GRU model, authors explained that the computational cost was increased with no static features due to the window size increasing from 87 to 298. How about window sizes in other models? Does it remains same in with/without static features?

This question relates to the previous question 2. All Window sizes are listed in table 6. To answer your question, each window size happens to be different. But for the other 2 model with a batch size of 256 the window size is smaller when static features are used.

6) It is incorrect to express "Across all models, meteorological features, particularly precipitation and average precipitation, consistently exhibited significant positive impacts on discharge prediction in lines 620." The temperature shows a negative relationship with discharge variation.

The soil temperature was grouped within the soil attribute class and not within meteorology, hence the sentence is correct within the context of our paper. However, since we shortend the sensitiviity analysis with regard to the other reviewer, the issue is solved.

7) It is suggested to compare the predictive discharge from three models in one gauge station of one figure in the Appendix. It seems that predictive performance of three models are different in the same gauge. Does it relate to the spatial distribution of gauges?

We added hydrographs for 3 different gauges in Appendix A4. The following sentence was added "Appendix A4 presents a comparison of the simulated hydrographs for the same basin. Consistent performance trends are observed across all models, with either poor or high performance in the same basin. However, one plot exhibits mixed performance, where both LSTM and GRU models perform well, while the CNN model shows poor performance. Notably, this is the only validated catchment where such a strong discrepancy is observed."(L725-728)

---

## Author Response (AR2)

**Review of HESS Manuscript**

"Neural networks in catchment hydrology: A comparative study of different algorithms in an ensemble of ungauged basins in Germany"

Dear Editor,

Please find attached the second review of the manuscript.

**Major comment 1**

In sections 2.5.1 and 2.5.2, you explain that after the dense layer of both the LSTM and the GRU, you applied a sigmoid. Is there a reason why this was done? This is not a usual approach, and existing benchmarks such as Kratzert2019 for CamelsUS, Lees2022 for CamelsGB and Loritz2024 for CamelsDE, do not apply this. A sigmoid is going to remove negative values but is also going to restrict high values. This might be the reason why the minmax transformation was working better (mentioned in the first revision), because you were restricting the training data to a 0-1 range.

With the sigmoid you cannot go above 1. Therefore, even during validation and testing, there is a structural restriction that your model cannot produce values larger than the maximum value in training. Also, the sigmoid will saturate the output of the model for large values, so even if the lstm/gru is trying to go higher, the sigmoid will cut the value short, and will disproportionally constrict higher values to a smaller range (due to the gradient of the sigmoid in high values of x). This is a major structural deficiency for the models.

In the course of finalizing our modeling setup, we systematically evaluated multiple normalization and activation strategies. First, we tested both standardization (as applied by Kratzert et al., 2019) and MinMax scaling for each of the three models under investigation (CNN, LSTM, and GRU). As shown in Figure 1, MinMax scaling consistently yielded significantly higher average KGE values. While we did not conduct a detailed root-cause analysis to explain why standardization performed less favourably in our study, the superior performance of MinMax scaling, documented in Figure 1, provided a clear motivation to pursue this approach.

In parallel, we also evaluated four different output-layer activation functions for the LSTM. According to Table 1, the sigmoid activation produced the highest max KGE results, slightly outperforming alternative functions such as Leaky ReLU, Softplus and Linear. Given these findings and in the interest of consistency across the recurrent architectures, we employed sigmoid activations in both the LSTM and GRU models without conducting a separate activation function analysis for the GRU model. However, as can be observed in Table 1, the performance differences between sigmoid and leaky ReLU were relatively small, with leaky ReLU even performing slightly better for the GRU model. Based on the concerns you raised regarding the restrictive nature of the sigmoid function—such as its inherent limitation on output range and its saturation effect for larger values—we acknowledge that leaky ReLU might have been a more suitable choice. Nonetheless, our decision to use the sigmoid activation function was driven by the fact that our MinMax-scaled data ranged between 0 and 1. Additionally, when using leaky ReLU, we observed instances of negative discharge predictions, which are physically implausible and undesirable in our specific hydrological context. Given this, the sigmoid activation function appeared to be a reasonable choice despite its structural limitations.

We acknowledge, however, the deficiency when using sigmoid combined with a MinMax Scaling approach. Future research may wish to explore alternative approaches—such as reverting to Leaky ReLU or employing specialized output transformations—to enable the model

to capture very high discharge events more accurately. We have added a recommendation to this effect in the Discussion section, noting the trade-off between potentially negative predictions and the ability to represent the full dynamic range of discharges.

In summary, although we concur that a final sigmoid layer could impose structural limitations on predicted discharge extremes, our empirical evaluations showed that this configuration, in combination with MinMax scaling, resulted in the most robust performance across the training and test datasets used in this study. We have incorporated an explicit recommendation in the manuscript for future work to revisit these design decisions, especially concerning the choice of activation function.

Thank you for highlighting this point, as it underscores the importance of scrutinizing how normalization and activation function choices can impact model performance and interpretability.

Figure 1 Comparison of Min-Max Scaling versus Standardization with Respect to Activation Function: Each boxplot represents the distribution of the mean Kling-Gupta Efficiency (KGE) across all catchments for 10 iterations.

Figure 2 Comparison of different activation functions Each boxplot represents the distribution of the mean Kling-Gupta Efficiency (KGE) across all catchments for 10 iterations.

Table 1 Optimal Activation Functions for Various Metrics Across Different Models

| CNN  | Median     |        | Mean       |        | Max        |        |
|------|------------|--------|------------|--------|------------|--------|
|      | leaky_relu | 0.7460 | leaky_relu | 0.7500 | leaky_relu | 0.7893 |
| GRU  | leaky_relu | 0.7537 | linear     | 0.7442 | leaky_relu | 0.7703 |
| LSTM | sigmoid    | 0.7503 | leaky_relu | 0.7214 | sigmoid    | 0.7688 |

**Major comment 2**

About the statistical-significance tests suggested in the first review process, the authors indicated in their response that:

"...However, as it is not our intention to claim at this point that any of tested models is better or worse, we decided to leave this test for future work."

I do not agree with this point. The title of the paper states "...A comparative study of different algorithms..." and section 3.1 directly compares the models. Moreover, you state conclusions as "Despite the general use of LSTM models, this study demonstrated that CNN models offer advantages in terms of performance and runtime for time series prediction." Therefore, I think you are testing which models are better or worse, and to test if the reported differences are significant or just due to random initialization, would greatly benefit the paper. The ensembles are created after all the hyperparameter tunning steps, and given the reported times to train each model, I think would be feasible to do this.

As suggested by the reviewer, we have added a section demonstrating the statistical robustness of the values in line 387-400

**Minor comments**

Line 8: "dynamic input features" is a more common term than "non-static input features".

**Changed as proposed**

Line 26. Cite LSTM paper, (Hochreiter and Schmidhuber, 1997).

**Changed as proposed**

Line 44: Change "one model fits all" to "regional".

**Changed as proposed**

Line 62: I would not say that LSTM incurs at tremendous computational costs. One can train for the whole CAMELS-US or CAMELS-GB in around 5 hours, if one has a normal GPU. It is for sure more than a FFNN, but considering other deep learning applications, it is quite affordable, and even comparable to training ~600 conceptual models.

**Changed as proposed**

Line 60-70: This is more of a personal style, but as a suggestion, in a technical report one should avoid adjectives such as "distinguished" and "renowned". Also, at the beginning of the introduction, you mention "paramount importance". It is better to just state facts. Again, this is personal style.

**Changed as proposed**

Line 112: Change Moreover to Furthermore, because the previous connection in line 109 was Moreover.

**Changed as proposed**

Line 134: I would remove "remarkably".

**Changed as proposed**

Line 166: The (Hochreiter and Schmidhuber, 1997) citation in this line is not consistent. What part of the phrase are you citing? If it is "have been extensively discussed in prior research" then cite it after part, otherwise you should remove it.

**Changed as proposed**

Line 259: Change to "a multiple of...".

**Changed as proposed**

Line 353: Rephrase "a testament to the proficiency of these artificial model"

**Changed as proposed**

Line 527: Which notable differences are you referring to?

This sentence was a fragment of a previous revision, where the sensitivity analysis included static features. It is now removed.

Appendix A: Units of discharge are mm/day. Also, all the figures in this appendix have the same name! The names should be different and indicate the details of the figure.

Changed as proposed

**Review of HESS Manuscript**

"Neural networks in catchment hydrology: A comparative study of different algorithms in an ensemble of ungauged basins in Germany"

Dear Editor,

Please find attached the second review of the manuscript.

**Major comment 1**

For the metrics provided in Table 5, it seems the differences are not obvious among the three methods. For example, the mean of KGE in the case of with batch size of 256 and +SF features is 0.8, 0.78 and 0.77 for CNN, LSTM and GRU respectively. In addition, some parameters such as window size shown in Table 6 are different. It is difficult to interpret the models' performance based on their algorithms and make a comparison. Could authors explain more the different behaviours from three methods performance like the model structure?

We added a paragraph analysing the influence of the window size towards the model performance. (line 384-400)

**Major comment 2**

All models failed to make good prediction of lower flows (Q1, Q2 and even Q3 in Figure 7). Authors tried to avoid a 'smooth' prediction using RMSE which capture the mean variability in the training dataset but instead using the KGE as lose function. Although the correlations, deviations and means of observations and simulations are considered in the KGE, the predictions produced high variability ratios. Authors explained that "This phenomenon may be attributed to a bias in the KGE towards elevated flows, thereby inadequately penalising inaccuracies in lower flow predictions in Lines 467-468". Could authors make a clearer explanation for that? In addition, I was wondering whether these behaviours are related to the training datasets of the 54 catchments, which may include a large part of high flows but limited with low flows.

A clearer explanation of this issue has been provided in lines 501–505. Regarding the second part of the question, Figure 1 presents a histogram comparing the discharge distributions of the training and test datasets. The test dataset exhibits a slightly higher proportion of very low discharge values than the training dataset, suggesting a potential underrepresentation of low-flow conditions in the training set. In contrast, the distributions for moderate and high discharge values are largely similar between the two datasets. Although the highest discharge values are cut off in the histogram, the QQ plot (Figure 2) shows that the test dataset contains higher absolute discharge values than the training dataset for highest flows.

Figure 3 Comparison of Discharge Distributions in Training and Test Datasets.

Figure 4 Quantile-Quantile (QQ) Plot Comparing Training and Test Discharge Distributions.

3) For the run time in Section 3.2, does the measured runtime including the training time or only the online prediction time?

The runtime of the model specifically refers to the training duration. This has been clarified in line 417.

4) It is not clear that whether discharge changes (%) in different scenarios were found in all test catchments or the averaged changes of the test catchments.

In line 548 we stated "The newly predicted discharge values were then systematically averaged over both time and all catchments" However, to provide further clarification, we have added the following sentence in line 552 – 553 "The results of this analysis are shown in Figure 10 representing the mean percentage change in discharge, calculated by averaging over all daily predictions and across all 35 catchments."

5) Authors made a detailed analysis among the three models in terms of different metrics, segment assessment, and model sensitivity. However, it is not clear for readers to understand the differences in results and the more detailed explanations (e.g., comparative tests or references) and discussions are suggested to include in results.

We do not fully understand the reviewer's question and would appreciate additional clarification to ensure we address the concern appropriately.

---

## Author Response (AR3)

**Review of HESS Manuscript**

"Neural networks in catchment hydrology: A comparative study of different algorithms in an ensemble of ungauged basins in Germany"

**Report 1**

Thanks for authors' efforts on replying to the comments and making revisions. My comments are as follows.

**Specific comments:**

Authors made a detailed analysis among the three models in terms of different metrics, segment assessment, and model sensitivity. However, it is unclear for readers to understand the significant mechanisms in the three methods which contribute to varied performances. The conclusions stated that CNN model offered superior performance, LSTM model exhibited superior generalization capabilities across the entire spectrum of flow data, but the GRU model showed a promising balance between predictive accuracy and computational demand. Are these conclusions consistent with other studies, or only valid in this study area? I suggest to add a part to discuss the reasons causing the differences and limitations in the three methods as well as the suggestions for future research.

We thank the reviewer for their insightful comment. In response, we have thoroughly revised the conclusions to address the concerns raised. The updated version (L607-673) now explicitly discusses the underlying mechanisms that contribute to the differences in model performance. Furthermore, we have incorporated a critical comparison with existing literature to assess the generality of our findings. We also added a part, discussing the reasons behind the observed performance variations, the limitations of each method, and suggestions for future research directions.

**Dear Editor,**

I am attaching the third review of the manuscript.

**Comment 1: About the Min-Max scaler.**

To clarify, in my previous comment I was not criticizing the Min-Max transformation, I just indicated that probably this is the reason the sigmoid activation yields to better results, because both the target and the simulated values are mapped to a 0-1 space.

If you want to keep using Min-Max that is your decision. However, I believe there is a problem with the test you are conducting to justify this decision (Fig 1 on the author's response), where you show that the Min-Max works much better than the StandardScaler. There is no logical reason to have such a significant drop in performance when you use the StandardScaler. There are multiple benchmarks (Kratzert2019, Less2021, Loritz2024) that have used the StandardScaler, and none of them present such bad performances.

Moreover, in the CAMELS-DE study (Loritz et al., 2024) the authors run a LSTM that was trained on daily data, using the StandardScaler transformation and without a sigmoid activation. The authors report the results for 93 catchments in Hesse (the same region you are studying) and according to their results, the median NSE for the basins in Hesse was close to 0.88. Therefore, I do not think the results you presented in Fig1 of the author's response are correct. You can still use the min-max scaler, and keep the results of the articles as you did. This would not change the message of the paper. However, you should eliminate the parts where you indicate that the MinMax scaler was chosen because it gave better results, because the results you presented in Fig1 of the author's response are not consistent with existing literature. For example, I would suggest eliminating this sentence:

Line 120: Subsequently, the two data sets were normalised by employing the a min–max scaling method, with a range of [0,1] chosen as the boundaries. This method was favoured over the standardization approach employed by Kratzert et al. (2019a), as it consistently yielded superior predictive performance across all models utilized in the study. Reference:

Loritz, R., Dolich, A., Acuña Espinoza, E., Ebeling, P., Guse, B., Götte, J., Hassler, S. K., Hauffe, C., Heidbüchel, I., Kiesel, J., Mälicke, M., Müller-Thomy, H., Stölzle, M., & Tarasova, L. (2024). CAMELS-DE: Hydro-meteorological time series and attributes for 1582 catchments in Germany. Earth System Science Data, 16(12), 5625–5642. <a href="https://doi.org/10.5194/essd-16-5625-2024">https://doi.org/10.5194/essd-16-5625-2024</a>

**We thank the reviewer for their thoughtful and detailed comments.**

In contrast to our study, Loritz et al. (2024) and Kratzert et al. (2019a) did not incorporate categorical static features requiring explicit encoding. Therefore while using label encoding standardization might produce misleading scaled values, since the values are not normally distributed. We observed that two input features (precipitation, catchment size), as well as the target variable (discharge), exhibit strong positive skewness. Standardization assumes symmetric distributions and can be destabilized by extreme values, whereas MinMax scaling bounds all inputs within [0,1], promoting training stability, especially in networks using sigmoid activations. We have revised the manuscript to clarify that the choice of MinMaxScaler was empirical and dataset-specific, and have removed any generalized claims regarding its general superiority.

According to the reviewers suggestion we removed "This method was favoured over the standardization approach employed by Kratzert et al. (2019a), as it consistently yielded superior predictive performance across all models utilized in the study." and added Line 118-119 "The choice of this scaling method was made empirically based on observed performance in our dataset and model configuration."

Comment 2. About using a sigmoid at the end of the pipeline.

You are justifying using a sigmoid based on Fig1 of the response you gave. Again, if you want to keep using a sigmoid that is your decision, and probably the message of the article will not be affected. However, I do not think that the KGE metric, in which you based your decision, is the best for this case. The sigmoid will saturate in high values, and therefore the difference between the models with different activations will be seen in the highest peaks. Therefore, a metric as the KGE that gives an overview of the overall performance will probably not summarize the saturation problem caused by the sigmoid. If you want to see differences, you should focus on the highest peaks. Again, this will probably not change the points made on the paper, but you should consider that the explanation you are given can be biased by the metric you are reporting.

Even though in line 610 you are speaking clearly about the limitation given by the sigmoid:

"While the sigmoid activation function provided stable performance, its combination with Min–Max scaling constrained discharge predictions. Employing LeakyReLU could allow for greater flexibility in discharge predictions, albeit with the trade–off of potential negative values." I would suggest emphasizing this point a bit further. The sigmoid activation is artificially decreasing high flows and imposing a structural constraint that you cannot go above what you saw in training. You can keep using the sigmoid, and the message of the paper will probably still be the same, but you should state that other configurations should be preferred in practical cases, especially if one is interested in predicting extreme discharges (e.g, flood forecasting).

We thank the reviewer for highlighting the potential bias introduced by using KGE to evaluate model performance under sigmoid activation. We agree that sigmoid activations can theoretically induce saturation effects at both low and high flow extremes. However, as shown in our flow-segment analysis (Figure 8), the models demonstrated robust predictive skill primarily for the highest flow quartile (Q4), while performance for low and mid-range flows (Q1–Q3) was consistently poor across all architectures (KGE-metrics). This indicates that, despite the bounded nature of the sigmoid output, our models retained the ability to capture peak flow dynamics effectively, whereas low-flow conditions presented a greater challenge. Given that our analysis explicitly addressed the highest peaks, the reviewer's concern is not entirely clear to us.

**We added Line 656-661:**

"Certain design choices and limitations must be acknowledged. Both recurrent models (LSTM and GRU) constrained outputs to non–negative discharges within the training data range using sigmoid activation and min–max normalization. This constraint ensures physically plausible predictions but restricts extrapolation beyond maximum observed flows. This saturation effect may attenuate extreme flood peaks, limiting the model's extrapolation capacity. For practical applications requiring accurate flood forecasting (primarily focusing on high discharge), alternative activation functions such as LeakyReLU, which allow unbounded outputs, may offer greater flexibility and should be considered in future model designs."

Other minor comments for the article:

Line 26: Modify to: As demonstrated by Kratzert et al. (2019a), an artificial neural network (ANN) model, namely Long Short–Term Memory (LSTM) network (Hochreiter and Schmidhuber, 1997), has shown unprecedented accuracy in PUB.

Changed as proposed by the reviewers. The Text reads now: "As demonstrated by Kratzert et al. (2019a), an artificial neural network (ANN) model, namely Long Short–Term Memory (LSTM) network (Hochreiter and Schmidhuber, 1997), has shown unprecedented accuracy in PUB"

Line 34: DOI to Ghimire et al. (2021) is not working.

We could not find any issues with this DOI.

Line 120: I would suggest removing this sentence (see first comment).

Removed as proposed by the reviewer.